



# Major processes of the dissolved cobalt cycle in the North and equatorial Pacific Ocean

Rebecca Chmiel[1,2], Nathan Lanning[3], Allison Laubach[4], Jong-Mi Lee[4], Jessica Fitzsimmons[3], Mariko Hatta[5], William J. Jenkins[2], Phoebe J. Lam[4], Matthew McIlvin[2], Alessandro Tagliabue[6], and Mak A. Saito[2]

[1] MIT/WHOI Joint Program in Oceanography/Applied Ocean Science and Engineering, Woods Hole, MA, 02543, USA
[2] Department of Marine Chemistry and Geochemistry, Woods Hole Oceanographic Institution, Woods Hole, MA, 02543, USA
[3] Department of Oceanography, Texas A & M University, College Station, TX, 77843, USA
[4] Department of Ocean Sciences, University of California Santa Cruz, Santa Cruz, CA, 95064, USA
[5] Institute of Arctic Climate and Environmental Research, Japan Agency for Marine-Earth Science and Technology, Yokosuka, Japan
[6] School of Environmental Sciences, University of Liverpool, Liverpool, L3 5DA, UK

*Correspondence to*: Mak A. Saito (msaito@whoi.edu)

**Abstract.** Over the past decade, the GEOTRACES and wider trace metal geochemical community have made substantial
contributions towards constraining the marine cobalt (Co) cycle and its major biogeochemical processes. However, few Co speciation studies have been conducted in the North and equatorial Pacific Ocean, a vast portion of the world's oceans by volume and an important endmember of deep thermohaline circulation. Dissolved Co (dCo) samples, including total dissolved and labile Co, were measured at-sea during the GEOTRACES Pacific meridional transect (GP15) along the 152º W longitudinal from 56º N to 20º S. Along this transect, upper ocean dCo was linearly correlated to dissolved phosphate (slope
= 82 ± 2 µM:M) due to phytoplankton uptake and remineralization. As depth increased, dCo concentrations became increasingly decoupled from phosphate concentrations due to co-scavenging with manganese oxide particles in the mesopelagic. The transect revealed an organically-bound coastal source of dCo to the Alaskan Stream associated with low salinity waters. An intermediate-depth hydrothermal flux of dCo was observed off the Hawaiian coast at the Loihi Seamount, and the elevated dCo was correlated to estimated xs³He at and above the vent site; however, the Loihi Seamount likely did not
represent a major source of Co to the Pacific basin. Elevated concentrations of dCo within oxygen minimum zones (OMZs) in the equatorial North and South Pacific were consistent with the suppression of oxidative scavenging, and we estimate that future deoxygenation could increase the OMZ dCo inventory by 13–28 % over the next century. In North Pacific Deep Water (NPDW), a fraction of elevated ligand-bound dCo appeared protected from scavenging by the high biogenic particle flux in the North Pacific basin. This finding is counter to previous expectations of low dCo concentrations in the deep Pacific due to
scavenging over thermohaline circulation. Compared to a Co global biogeochemical model, the observed transect displayed more extreme inventories and fluxes of dCo than predicted by the model, suggesting a highly dynamic Pacific Co cycle.





# 1 Introduction

Cobalt (Co) is a necessary inorganic micronutrient for many phytoplankton and other forms of marine life but is one of the scarcest essential bioactive metals in the surface ocean. Many cyanobacteria, such as *Synechococcus* and *Prochlorococcus*, have an absolute requirement of Co (Saito et al., 2002; Sunda and Huntsman, 1995), while other phytoplankton require vitamin $B_{12}$ for methionine synthesis, which contains Co as a metallic cofactor (Bertrand et al., 2013). Many metalloenzymes also use Co as a cofactor, including some forms of eukaryotic carbonic anhydrases which can substitute Co for zinc (Zn) (Kellogg et al., 2020; Price and Morel, 1990; Sunda and Huntsman, 1995). The uptake of Co by marine plankton coupled with the scavenging of cobalt onto oxide particles in the mesopelagic ocean gives dissolved Co (dCo) a "hybrid-type" geochemistry, characterized by both nutrient uptake and scavenging process (Bruland and Lohan, 2003; Noble et al., 2008), which contributes to Co's short residence time in the water column and keeps dCo at picomolar concentrations in the open oceans (Hawco et al., 2018; Saito and Moffett, 2002). Co has been predicted to be a co-limiting nutrient in some regions of the ocean, particularly areas with low Zn concentrations (Browning et al., 2017; Moore et al., 2013; Saito et al., 2002). Some of Co's biological influence is through vitamin $B_{12}$, which has been found to be a co-limiting nutrient in regions like the Antarctic coastal seas (Bertrand et al., 2007; Gobler et al., 2007; Sanudo-Wilhelmy et al., 2006).

Major sources of Co to the marine environment include rivers and estuaries, coastal sediments, hydrothermal inputs, and dust deposition (Dulaquais et al., 2014a; Hawco et al., 2016; Noble et al., 2017). Major sinks of Co primarily require dCo to be transformed to the particulate phase (pCo) via uptake by phytoplankton or lithogenic scavenging before it is removed from the water column by particle export. Dissolved Co is principally scavenged to manganese (Mn) oxides via a microbially mediated co-oxidation (Hawco et al., 2018; Moffett and Ho, 1996; Noble et al., 2017). Internal cycling processes act as major sources of dCo from particulate matter via biomass remineralization and the build-up of dCo in oxygen minimum zones (OMZs). Dissolved Co tends to be elevated in OMZs due to the combined processes of biomass remineralization below the photic zone, suppressed Mn oxide scavenging in the anoxic environment, remobilization of scavenged Co via Mn oxide reduction, and reduced ventilation and mixing of the water mass (Hawco et al., 2016; Noble et al., 2017). Inputs of dCo from OMZs have been shown to be higher than previously expected, representing ~27 % of the total marine Co flux (Hawco et al., 2018). Comparatively, the hydrothermal and aeolian dust fluxes are predicted to be < 3 % of the total Co source flux (Hawco et al., 2018; Swanner et al., 2014).

Speciation determines the bioavailability of dCo for its major loss processes: uptake by phytoplankton and scavenging by Mn-oxidizing bacteria. Dissolved Co is present in both a labile, "free" Co(II) speciation and a ligand-bound, "complexed" speciation. Most if not all organically complexed dCo is predicted to be Co(III) due to its high binding stability with ligands. Ligand-bound Co(III) is a strongly inert fraction of the dCo pool with a conditional stability constant > $10^{16.8}$ (Saito et al., 2005). Organic Co(II)-ligand bonds are comparatively less stable since ligands have a higher affinity to divalent cations that are more abundant than Co such as Ni(II), making Co(II) ligand concentrations negligent (Baars and Croot, 2014). Labile dCo is considered to be more bioavailable to phytoplankton and oxidizing bacteria than the relatively inert ligand-bound dCo, but





there is some evidence that cyanobacteria are able to produce extracellular organic ligands to stabilize ligand-bound dCo for uptake as a nutrient and to prevent its loss via scavenging (Bown et al., 2012; Saito et al., 2005).

The efforts of the GEOTRACES and wider biogeochemical community have greatly increased our understanding of the marine Co cycle on a basin-wide scale over the past decade, including in the Atlantic, Arctic, Southern Ocean, South Pacific, and Mediterranean Sea (Bown et al., 2011, 2012; Bundy et al., 2020; Dulaquais et al., 2014a, 2014b, 2017; Hawco et

al., 2016; Noble et al., 2017). However, the distribution, sources, and sinks of marine Co have not been widely characterized in the North and equatorial Pacific, leaving a vast section of the ocean understudied with regards to Co speciation and biogeochemical dynamics. The North Pacific Ocean is the largest ocean basin with the oldest, least ventilated deep-water mass that serves as the end-member for deep thermohaline circulation. Understanding the major biogeochemical features of the Co cycle in this basin, particularly within the Pacific deep endmember, is important to our understanding of the global Co cycle.

In this study, we examine the distribution of both total dissolved and labile dCo along the GEOTRACES GP15 Pacific meridional transect through the North and equatorial Pacific Ocean. This paper characterizes the major features present in the dCo GP15 transect and compares them to similar features found in other ocean basins where a higher density of dCo measurements have been reported. The observed dCo transect is then compared to output from a global biogeochemical Co model to assess and contextualize dCo features along the transect. This study area was influenced by many relevant sources

and sinks within the Co cycle, including coastal and riverine fluxes, OMZs, hydrothermal vents, oligotrophic subtropical gyres, and the aged deep Pacific endmember, providing an ideal snapshot into the Pacific Co cycle and a representative model of global Co biogeochemical processes.

## 2 Methods

### 2.1 Study area and dissolved trace metal sample collection

North and equatorial Pacific seawater samples were collected on the GP15 expedition as part of the U.S. GEOTRACES program (RR1814 and RR1815; September 18–November 24, 2018). Dissolved seawater was sampled along a transect that mostly followed the 152º W meridian from 56º N to 20º S, traveling from the Gulf of Alaska to Hawaii (leg 1), and on to Tahiti, French Polynesia (leg 2; Fig. 1). Trace metal dissolved seawater samples were collected using a 24-bottle trace metal clean "GEOTRACES Carousel" (GTC) equipped with 12 L GO-Flo bottles (General Oceanics) on a titanium frame

and a Kevlar cable, as described in Cutter and Bruland, 2012. Seawater from the GO-Flo bottles was subsampled in a trace metal clean van under positive pressure. Surface water samples (~2 m depth) were collected from a trace metal clean towfish and pump while arriving at each station. Sampling locations included 25 full-depth stations and 10 demi stations where only the upper 1000 m was sampled. At station 11, only a surface towfish sample was collected due to rough weather.

Dissolved Co subsamples were filtered using a 0.2 µm Acropak capsule and stored until analysis at 4 ºC in 60 mL

LDPE bottles (Nalgene) that had been soaked for 1 week in Citranox, an acidic detergent, rinsed with Milli-Q water (Millipore),





soaked for 2 weeks in 10 % trace metal grade HCl (Optima), and rinsed with lightly acidic Milli-Q water (< 0.1 % HCl). All Co values presented here were analyzed at sea, with replicates of all samples taken for dCo analysis in the laboratory, if needed.

Macronutrient samples were analyzed at sea within 24 hours of recovery by the Ocean Data Facility at Scripps Institution of Oceanography. The nutrient and hydrographic dataset can be found on the Biological and Chemical

Oceanography Data Management Office (BCO-DMO) website (leg 1: https://www.bco-dmo.org/dataset/777951; leg 2: https://www.bco-dmo.org/dataset/824867). Macronutrient concentrations reported here have been converted from units of µmol kg$^{-1}$ to µM by multiplying each sample by its calculated *in-situ* density.

The depth of the mixed layer was determined for each station as the first depth at which the difference between the potential density ($\sigma_\Theta$) and reference density ($\sigma_{ref}$) was greater than or equal to 0.125 kg m$^{-3}$, where $\sigma_{ref}$ is the $\sigma_\Theta$ at 10 m depth

(Bishop and Wood, 2009; Ohnemus et al., 2017). *In-situ* density, potential density, and O$_2$ solubility used to calculated apparent oxygen utilization (AOU) were determined via Gibbs Seawater (GSW) Oceanographic Toolbox in Python (https://teos-10.github.io/GSW-Python/) (Mcdougall and Barker, 2011).

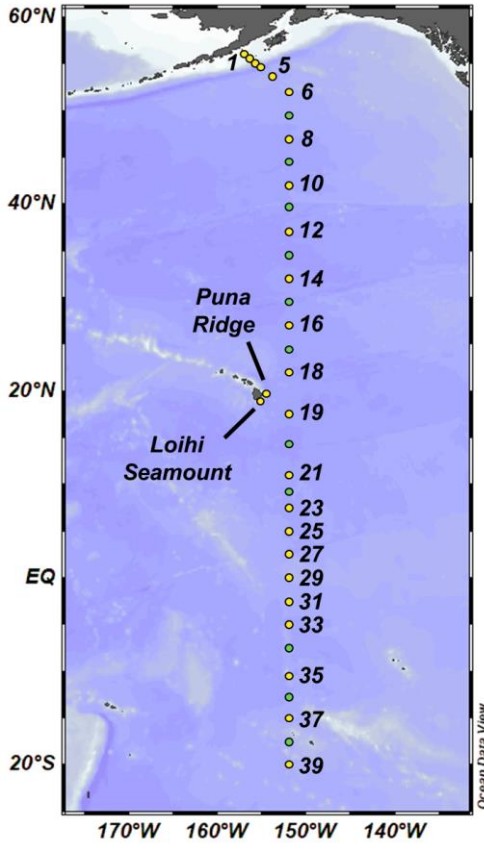

**Figure 1: GEOTRACES GP15 expedition track. Leg 1 (RR1814, Sept–Oct 2018) spanned between the Alaskan coast and Hawaii, and Leg 2 (RR1815, Oct–Nov 2018) spanned between Hawaii and -20° S. Yellow points represent full depth stations and green points represent demi stations where only the upper 1000 m was sampled. At Station 11, only a surface towfish (~2 m) sample was collected.**



## 2.2 Dissolved total and labile Co analysis

Dissolved cobalt was determined by cathodic stripping voltammetry (CSV) as originally described by Saito and

Moffett, 2001 and modified by Hawco et al., 2016 and Saito et al., 2010. Measurement occurred shipboard inside a trace metal clean plastic "bubble" within one week of sample collection using the Metrohm 663 VA and µAutolabIII systems equipped with a hanging mercury drop working electrode. The reagents used for this method included (1) 0.1 M dimethyglyoxime (DMG, Sigma Aldrich), (2) 0.5 M N-(2-hydroxyethyl)piperazine-N-(3-propanesulfonic acid) (EPPS, Sigma Aldrich) buffer, (3) 1.5 M $NaNO_2$ (Merck), and (4) a 25 pM $CoCl_2$ (Fisher Scientific) standard made in September 2018 for this expedition.

The DMG reagent was purified via recrystallization, and the EPPS and $NaNO_2$ reagents were run through treated Chelex-100 resin columns (BioRad) to remove trace metal contaminates. The Chelex was resin prepared as described in Price et al., 1988/89, and reagents were purified as described in Saito and Moffett, 2002.

For total dCo analysis, 0.2 µm filtered water samples were UV-irradiated in acid-washed quartz tubes for one hour using a water-cooled UV irradiation system (Metrohm 705 UV Digestor) to destroy natural ligand-bound Co complexes. Then,

11 mL of sample seawater was aliquoted into 15 mL acid-washed polypropylene vials, and 33 µL of the DMG reagent and 130 µL of the EPPS buffer was added to each vial. The samples were then processed on an autosampler (Metrohm 858 Sample Processor), which added 8.5 mL of the sample solution and 1.5 mL of the $NaNO_2$ reagent to a Teflon cup for electrochemical analysis. The mercury electrode performed a fast linear sweep from -1.4 V to -0.6 V at a rate of 5 V $s^{-1}$, which reduced the Co bound in the $Co(HDMG)_2$ complex from Co(II) to Co(0) and produced a Co reduction peak at -1.15 V with a height linearly

proportional to the amount of dCo present in the sample. A standard addition curve was generated for each sample analyzed with 4 automated additions of the 25 pM $CoCl_2$ standard. A type-I linear regression of the standard curve allowed for the calculation of the initial amount of Co present in the sample, as described in Saito and Moffett, 2001. Triplicate technical replicates were run on every sample to determine the precision of the method. Duplicate depths from different rosette casts were run when available.

Labile dCo analysis was identical to the total dCo method described above except the samples were not UV-irradiated prior to analysis to preserve the sample's ligands, and labile samples were allowed to equilibrate with the DMG and EPPS reagents overnight (~8 hours) before CSV analysis. This equilibration step allows time for the labile dCo to bind to the DMG reagent via competitive ligand exchange (K > $10^{16.8}$), and thus is defined relative to its thermodynamic behavior (Saito et al., 2005). These labile dCo measurements are not necessarily comparable to dCo measurements that are performed using non-

electrochemical methods, which are sometimes described as labile. In particular, inductively coupled plasma mass spectrometry (ICP-MS) measurements of dCo where samples are not UV-irradiated prior to analysis gives consistently different results than this electrochemical method, likely due to differences in cobalt complex adsorption and/or exchange kinetics with metal binding resins. As a result, ICP-MS labile Co measurements tend to be somewhat offset compared to electrochemical labile Co measurements (unpublished data). This offset is not yet well understood, but it indicates that the two





methods are measuring different pools of the dCo inventory, which should be considered before comparing dCo values analyzed with different methodologies.

Peak heights were determined by ElectroChemistry Data SOFTware (ECDSOFT) when possible, and manual measurement of peaks was used when peak detection was unsuccessful by the available software. This hybrid approach was employed because the most updated version of Metrohm software at the time, Nova 2.0, no longer supported manual peak-

picking of individual samples, and ECDSOFT did not always detect peaks from samples with low dCo concentrations. A pairwise sample t-test was performed to compare the differences between the manual analysis method and the digital ECDSOFT analysis method as determined by selected samples that were analyzed using both methods (n = 13, average difference = $0.9 \pm 2.7$ pM). The two peak height analysis methods were found to be statistically equivalent (t = 1.16, P = 0.27).

The entire dCo dataset generated on the GEOTRACES GP15 expedition can be accessed online at https://www.bco-

dmo.org/dataset/818383 (leg 1) and https://www.bco-dmo.org/dataset/818610 (leg 2).

## 2.3 Analytical blanks

Analytical blanks for each combination of reagent batches (a unique combination of DMG, EPPS, and $NaNO_2$ reagents) were measured to determine background Co contamination due to reagent impurities. Blanks were prepared in duplicate or triplicate with UV-irradiated surface seawater passed through a treated Chelex-100 resin bead column (Bio-Rad)

to remove metal contaminants, then UV-irradiated again to ensure no excess Co ligands were present. Averaged analytical blank values were subtracted from the measured Co values determined with the respective reagent batch.

For the 4 reagent batches used on GP15, the analytical blanks were found to be $9.44 \pm 0.56$ pM (n = 3), $9.68 \pm 0.40$ pM (n = 2), $0.04 \pm 2.14$ pM (n = 2), and $6.28 \pm 0.84$ pM (n = 2; Table 1). For the third reagent batch (0.04 pM blank), no blank value was subtracted from the determined Co values because the standard deviation was greater than the measured blank Co.

An analytical limit of detection for this method was calculated from the $9.44 \pm 0.56$ pM blank ($3 \times$ Blank Standard Deviation where n ≥ 3) to be 1.7 pM.

## 2.4 Intercalibration and internal standards

While processing samples at sea, duplicate GEOTRACES community intercalibration standards – either GSC2 or D1 – were run at least once every 2 weeks (Table 1). All intercalibration standards are stored acidified, so samples were carefully

titrated directly before analysis with negligible volumes of ammonia hydroxide ($NH_4OH$) until they reached a pH of 7.5–8.2, then were UV-irradiated and analyzed as a dCo sample. The D1 standard ($44.0 \pm 0.01$ pM, n = 2) was found to be within one standard deviation of the consensus value ($46.6 \pm 4.8$ pM), while the GSC2 standard ($82.8 \pm 2.9$, n = 4) was found to be slightly higher than was reported in (Hawco et al., 2016) ($77.7 \pm 2.4$, 6.2 % difference). Additional GSC2 standards were run in the laboratory in August to November 2019 with a new reagent batch, and displayed dCo concentrations of $80.2 \pm 6.2$ pM (n = 3).

No official community consensus for dCo in GSC2 currently exists.





**Table 1: Blanks, internal standard, and GEOTRACES community intercalibration standards measured at sea during Co analysis.**

| | dCo [pM] | n | Consensus |
|---|---|---|---|
| **Blanks[a]** | | | |
| Sep. 18, 2018 | 9.44 ± 0.56 | 3 | |
| Oct. 6, 2018 | 9.68 ± 0.41 | 2 | |
| Oct. 27, 2018 | 0.035 ± 2.1 | 2 | |
| Nov. 9, 2018 | 6.28 ± 0.84 | 2 | |
| | | | |
| **Internal Standards** | | | |
| Sep. 26 - Nov. 9[b] | 55.9 ± 3.9 | 18 | |
| Oct. 22 - Nov. 21[c] | 15.5 ± 3.4 | 7 | |
| | | | |
| **Intercalibration Standards** | | | |
| GSC - Shipboard | 82.8 ± 2.9 | 4 | 77.7 ± 2.4[d] |
| GSC - Laboratory[e] | 80.2 ± 6.2 | 3 | |
| SAFe D1 | 44.0 ± 0.0065 | 2 | 45.4 ± 4.7 |

[a] **Pacific seawater, chelexed to remove metals and UV-irradiated.**
[b] **Surface North Pacific seawater, collected September 26, 2018.**
[c] **South Atlantic seawater, collected on the CoFeMUG expedition, 2007.**
[d] **From Hawco et al., 2016. Not an official consensus.**
[e] **Analyzed August, 2019 to November, 2019.**

To validate the consistency of the instrument, an internal standard was analyzed at least once every other day Co samples were measured. Two internal standard batches were used during the expedition: the first created from UV-irradiated surface North Pacific seawater (dCo = 55.9 ± 3.9 pM, n = 18) and the second created from UV-irradiated South Atlantic water collected on the 2007 CoFeMUG expedition (GAc01; KN192-5, 2007) (dCo = 15.5 ± 3.4 pM, n = 7) (Table 1). To ensure Co analysis was consistent while transitioning between internal standards, both standard batches were measured during an overlap period of two 2 weeks (Oct. 22–Nov. 9, 2018).

Station 35 of the GP15 expedition was a crossover station with the GEOTRACES GP16 expedition in 2013, and was located at the same latitude and longitude as GP16 station 36 (10.5º S, 152º W). A comparison of their dCo profiles shows good agreement between the two stations (Fig. 2), although an independent t-test of the deep dCo values ($\geq 2000$ m) does show a significant offset (5 pM) between the two crossover station profiles (t = -4.9, P = $2.7 \times 10^{-5}$). The GP15 measurements showed a deep ($\geq 2000$ m) dCo signal of 24.0 ± 2.6 pM (n = 15), and the GP16 measurements showed a slightly higher deep dCo signal of 29.1 ± 3.5 pM (n = 19). The GP16 profile from 2013 also had a larger mesopelagic maximum between 500 and 1000 m than the GP15 profile, which may be due to seasonal or annual variability in the remineralization signal in the South Pacific Gyre. The 5 pM offset may be due to a small analytical offset or oceanographic variability, possibly related to the higher mesopelagic dCo concentrations observed in the GP16 profile. Although the two profiles have a small statistical difference at depth, their substantial overlap and similarity of geochemical features are good indications that our measurement of dCo on the GP15 expedition was robust and oceanographically consistent.





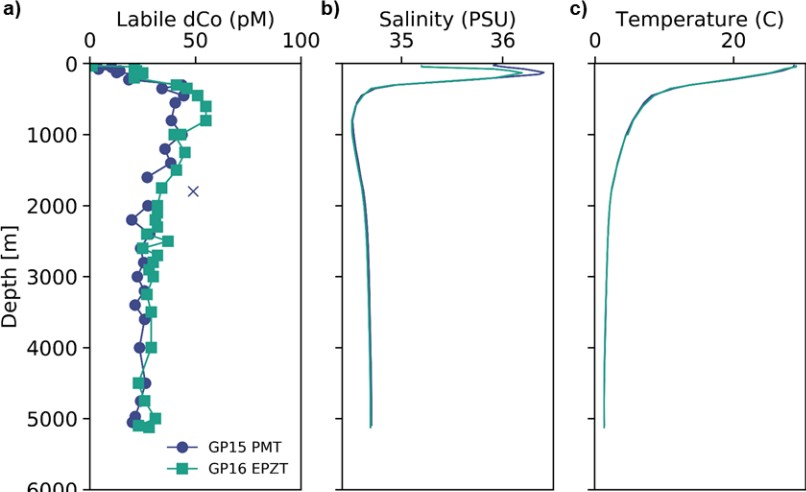

**Figure 2: (a) Dissolved Co, (b) salinity, and (c) temperature profiles from a crossover station between the GEOTRACES GP15 Pacific meridional transect (PMT, Nov. 2018; navy) and the GEOTRACES GP16 East Pacific Zonal Transect (EPZT, Dec. 2013; teal) at 10.5° S, 152° W. Outliers are marked with an X.**

## 2.5 Particulate metal and particulate organic carbon analysis

Samples for both particulate trace metal analysis and particulate organic carbon (POC) analysis were collected with dual-flow McLane Research in situ pumps (WTS-LV) at each full-depth station. Each McLane pump contained two filter holders: one outfitted with paired 1 μm quartz fiber (Whatman QMA) filters and 51 μm polyester prefilters (Sefar), and the other with paired 0.8 μm polyethersulfone (Pall Supor) filters and 51 μm polyester prefilters. Large particulate, total (LPT) samples were collected from the prefilters (> 51 μm). Small particulate, total (SPT) samples were collected from the QMA filters (1–51 μm) and Supor filters (0.8–51 μm), depending on sample type; SPT trace metal and lithogenic samples were collected from the Supor filters, while SPT POC samples were collected from the QMA filters.

Particulate trace metal samples, including particulate Co (pCo) and particulate Mn (pMn), were digested using hot refluxing with hydrofluoric and nitric acids and measured via high-resolution ICP-MS. Particulate aluminum (pAl) and titanium (pTi) concentrations were used as tracers for lithogenic particles and were used to calculate a lithogenic correction for pCo and pMn where the corrected values represent excess trace metals unaccounted for by a lithogenic source. The Co:Ti ratio between potential lithogenic endmembers was less variable (RSD ~27 %) than the Co:Al ratio (RSD ~64 %), so measured pTi was used for lithogenic Co correction together with a molar Co:Ti ratio of 0.00306, which was the median Co:Ti ratio of four potential lithogenic endmembers: upper and bulk continental crust (Taylor and McLennan, 1995), mid-ocean ridge basalt (Gale et al., 2013), and ocean island basalt 0.0306 (Hémond et al., 1994). For Mn, the Mn:Al ratio was less variable (RSD ~36 %) than the Mn:Ti ratio (RSD ~48 %) between lithogenic endmembers, so measured pAl together with a molar Mn:Al ratio of 0.0086, the median of the four lithogenic endmembers as referenced above, was used to correct for lithogenic Mn. Total particulate values were calculated by summing the SPT and LPT values.



POC samples were analyzed via combustion in an elemental analyzer after fuming with HCl to remove particulate
inorganic carbon. A particulate sampling and analysis procedure can be found in more detail in Xiang and Lam, 2020.

## 2.6 Excess ³He, dFe, and dMn analysis at the Loihi Seamount

The collection, analysis, and interpretation of helium (He) isotope and other noble gas samples along the GP15
transect and at the Loihi Seamount (station 18.6) has been presented by Jenkins et al., 2020. Samples of $^3$He, $^4$He, and Ne were
collected from Niskin bottles on the Ocean Data Facility (ODF) CTD rosette, and not from the trace-metal clean GEOTRACES
CTD carousel (GTC) that supplied the dCo samples. Briefly, He isotopic ratios and abundances were determined by
quantitative gas-extraction and mass spectrometry as described in Jenkins et al., 2019. Dissolved He and neon (Ne) saturation
anomalies were calculated as a function of temperature and salinity, and He isotope ratio anomalies were calculated relative
to a marine atmospheric standard. The excess $^3$He anomaly (xs$^3$He) was determined as the $^3$He unaccounted for by gas
saturation, surface air injection, and diapycnal mixing (Jenkins et al., 2019).

The CTD cast discrepancy between the noble gas samples (ODF) and the dCo samples (GTC) is of particular note at
the Loihi station because there was a clear offset in chemical composition between the two sampling casts, likely due to
currents pushing the rosettes to differing locations within the heterogeneous vent environment of the seamount (see Sect. 4.3).
The offset between the casts made a direct comparison between dCo and the xs$^3$He associated with hydrothermal sources in
the Loihi Seamount impossible. Instead, we used dissolved Fe (dFe) and dissolved Mn (dMn) samples taken from both the
GTC and the ODF casts to estimate a "potential $^3$He anomaly" from the GTC cast, assuming the dFe:xs$^3$He and dMn:xs$^3$He
ratios were consistent between the two casts. The ODF casts' trace metal:xs$^3$He ratios were used to infer the potential xs$^3$He
concentration profile from GTC dFe and dMn concentrations. This calculation is explained in detail in Sect. 4.3.

This paper presents dFe and dMn data determined by flow injection analysis (FIA) for the ODF cast and ICP-MS for
the GTC cast, as described in Jenkins et al., 2020. The two methods are in good agreement with each other, as shown in Figure
3 of Jenkins et al., 2020.

## 2.7 Pairing GP15 samples with existing radiocarbon data

To estimate sample conventional radiocarbon age, sample locations along the GP15 transect were matched with
existing radiocarbon data from the Global Ocean Data Analysis Project (GLODAP) v2, 2019 merged and adjusted data product
for the Pacific Ocean (https://www.glodap.info/index.php/merged-and-adjusted-data-product/) (Key et al., 2015; Olsen et al.,
2016, 2019). GP15 sample locations were paired with the closest GLODAP sample locations containing $\Delta^{14}$C data using their
latitudinal, longitudinal, and depth dimensions. Only GLODAP data points that were within 200 m depth and 500 km
latitude/longitude distance from the original GP15 sample location were accepted. The conventional radiocarbon age of each
deep sample (> 3000 m) was calculated with the equation:

$$\text{Radiocarbon Age} = -8033 \text{ yr} \times \ln(1 + (\Delta^{14}C + 67 \text{ ‰})/1000) \tag{1}$$





where 8033 is the inverse of the $^{14}C$ decay constant ($\lambda$; $1.21 \times 10^{-4}$ years$^{-1}$), and -67 ‰ is subtracted the $\Delta^{14}C$ value to correct for preformed $\Delta^{14}C$ in the North Atlantic such that the newly formed North Atlantic Deep Water (NADW) is reset to an age of ~0 years (Matsumoto, 2007). This radiocarbon age calculation has previously been used to calculate Co scavenging rates

via thermohaline circulation (Hawco et al., 2018), and does not correct for the increase in anthropogenic radiocarbon due to nuclear weapons testing, the change in the atmospheric $^{14}C$ due to increased use of anthropogenic fossil fuels via the Seuss Effect, or the mixing and ventilation of NADW with Antarctic Bottom Water (AABW) during ocean circulation. The lack of correction for ventilation during AABW formation is acceptable because most of the $\Delta^{14}C$ and dCo in the deep Pacific is derived from NADW, not AABW (Hawco et al., 2018). The radiocarbon age calculated here is not the "natural" or

"circulation" radiocarbon age and is only used to represent relative ages and ventilation of water masses in the deep Pacific Ocean.

## 2.8 Comparison to a cobalt biogeochemical model

The observational dCo concentrations from bottle samples along the GP15 transect were compared to the output of a published global Co biogeochemical model (Bundy et al., 2020; Richon and Tagliabue, 2021; Tagliabue et al., 2018). The Co

model is embedded within the BYONIC version of the PISCES-v2 biogeochemical model (Richon and Tagliabue, 2021), which includes the simulation and circulation of a variety of tracers including dissolved phosphate (dPO$_4$), nitrate, ammonium, silicic acid, Fe, copper (Cu), Zn, Mn and O$_2$, as well as resolving different phytoplankton, zooplankton and particle pools. The Co biogeochemical model (Tagliabue et al., 2018) incorporates dCo, scavenged Co, diatom-associated Co, nanophytoplankton-associated Co, small particulate organic Co and large particulate organic Co. External input of Co is via dust, rivers and

continental margins, with margin Co supply modulated by the overlying O$_2$ content. The model output was gridded on a 1º × 1º latitude/longitude grid with 33 depth boundaries: 0, 10, 20, 30, 40, 50, 60, 70, 80, 90, 100, 110, 120, 135, 150, 175, 200, 250, 300, 500, 700, 1000, 1250, 1500, 2000, 2500, 3000, 3500, 4000, 4500, 5000, and 5500 m. Gridded points along the GP15 transect were extracted from the annual mean global model results and compared to the observed GP15 transect. In order to appropriately compare the two datasets, the dCo, O$_2$, and dPO$_4$ observed concentrations along the GP15 transect were binned

on the same grid as the model (1º × 1º latitude/longitude with 33 depth boundaries), and the values of any observed data points that occupied the same bin were averaged.

## 2.9 Regression analysis

All linear regressions presented in this paper were made using a two-way (type-II) linear regression, with the exception of the standard addition curves used to calculate dCo concentrations (Sect. 2.2). Two-way regressions are more

suitable for stoichiometric analysis than one-way (type-I) regressions because they allow for error in the x parameter as well as the y parameter, and as such do not assume dependence between the x and y axes. The algorithm for the two-way regression



was written into MATLAB (lsqfitma.m) by Ed Peltzer circa 1995 (Glover et al., 2011), and was rewritten as a Python 3 function for this paper (available at https://github.com/rebecca-chmiel/GP15).

## 3 Results

### 3.1 Hydrographic setting

The GEOTRACES GP15 expedition (Fig. 1) investigated a cross section of the North and Central Pacific basin that contained a wide range of ocean biogeochemical processes, including the North and South Pacific low-nutrient gyres, shelf and riverine-associated sedimentary inputs along the Alaskan coast, hydrothermal vent inputs near the Hawaiian Loihi Seamount, and three oxygen minimum regions.

The GP15 transect crosses 5 major Pacific oceanographic provinces, including: (1) the Alaskan coast (56º N–55º N, stations 1–3), characterized by shallower depths, low salinity surface waters, and the coastal Alaskan Current (Reed, 1984; Stabeno et al., 1995); (2) the North Pacific subpolar gyre (54.6º N–44.5º N, stations 4–9), an area of cyclonic upwelling and a high nutrient, low chlorophyll (HNLC) region generally considered to be iron (Fe) limited  (Boyd et al., 2004; Tsuda et al., 2005); (3) the North Pacific subtropical gyre (42º N–17.5º N, stations 10–19), an anticyclonic oligotrophic gyre with low surface macronutrients and intermediate suboxic waters; (4) the equatorial Pacific (14.25º N–7.5º S, stations 20–34), characterized by equatorial upwelling, subsurface equatorial currents, and the cross sections of the Eastern Tropical North Pacific (ETNP) and Eastern Tropical South Pacific (ETSP); and (5) the South Pacific subtropical gyre (10.5º S–20º S, stations 35–39), an oligotrophic anticyclonic gyre that is generally macronutrient-limited, particularly for nitrogen (N) (Bonnet et al., 2008).

Three distinct Oxygen Minimum Zones (OMZs) were present in the GP15 transect: the tail ends of the ETNP and ETSP OMZs, around 5–15º N and 2–10º S respectively, and the intermediate waters of the North Pacific (Fig. 3c). OMZs are created by a combination of oxygen consumption, often via remineralization of organic carbon, and poor water mass ventilation that forms a "shadow zone" of sluggish, isolated water (Karstensen et al., 2008; Lavergne et al., 2017). In the ETNP and ETSP, shadow zones are created by Ekman upwelling along Eastern Pacific margins of North and South America (Czeschel et al., 2011; Karstensen et al., 2008); these OMZs are wider and more depleted in $O_2$ closer to the American continents where they are formed (Hawco et al., 2016; Stramma et al., 2008), and the GP15 transect captured a cross-section of the less $O_2$-depleted tailings of the OMZs. In contrast, the massive North Pacific OMZ is created by the abyssal overturning of North Pacific Deep Water (NPDW) traveling northward via thermohaline circulation. When the dense NPDW reaches the North Pacific, it is upwelled to mid-depth waters (< 2500 m) and returns southward along intermediate isopycnal boundaries, creating a shadow zone of low circulation in the above waters (1000 m–2500 m) (Lavergne et al., 2017). The combination of the shadow zone and the remineralization of organic carbon from upwelling-fueled phytoplankton blooms in the North Pacific subpolar gyre (Nishioka et al., 2020; Sarmiento et al., 2004) creates a large OMZ that stretches across the North Pacific transect at intermediate depths. The three Pacific OMZs alter the redox state and biogeochemistry of trace metals contained within them.

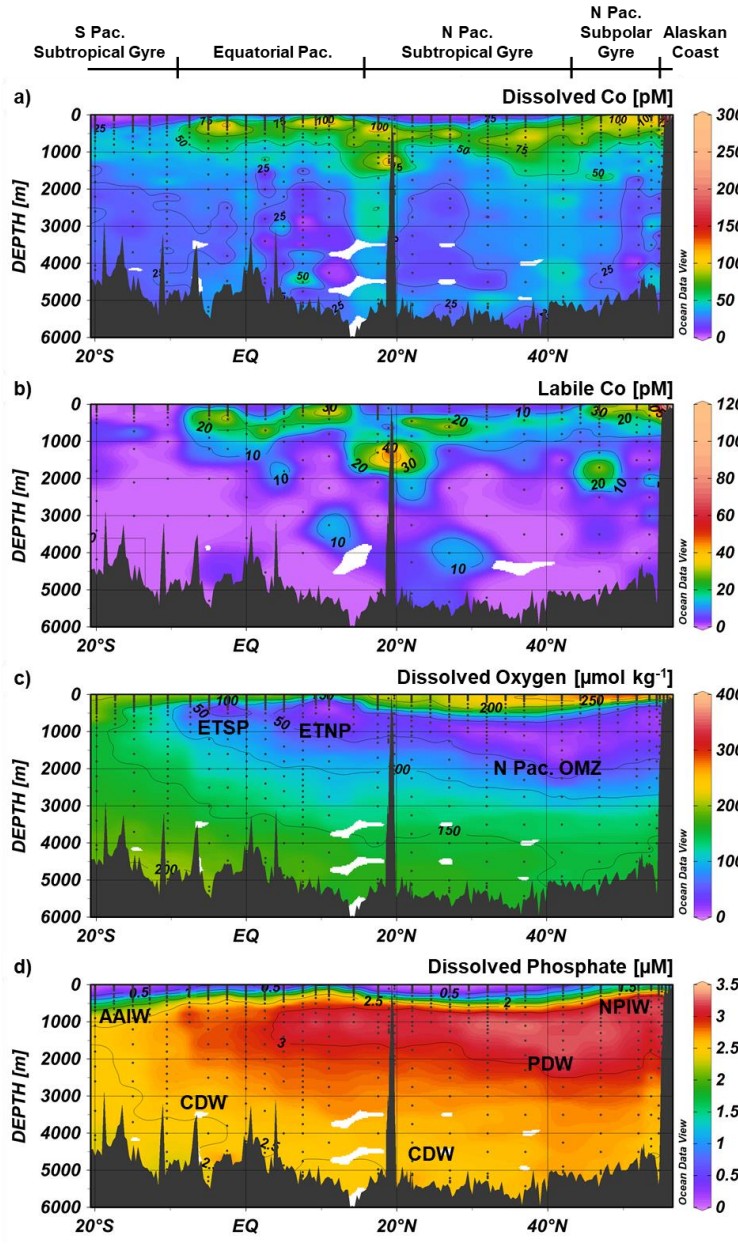

**Figure 3: Dissolved Co (a), labile dissolved Co (b), dissolved oxygen (c), and dissolved phosphate (d) transects along the GP15 cruise track. The Eastern Tropical North Pacific (ETNP), Eastern Tropical South Pacific (ETSP), and North Pacific OMZs are labelled, as well as Antarctic Intermediate Water (AAIW), Circumpolar Deep Water (CDW), Pacific Deep Water (PDW), and North Pacific Intermediate Water (NPIW). Weighted-average gridding is used to interpolate between data points.**

### 3.2 Dissolved cobalt distributions in the Pacific Ocean

During the GP15 expedition, 1024 samples were analyzed at sea for Co, including 715 total dCo measurements (hereafter simply dCo) from 36 stations and 309 labile dCo measurements from 22 stations. The resulting longitudinal transect



(Fig. 3, Fig. 4) represents the second largest Pacific dataset of dCo speciation measurements, after the GEOTRACES GP16 dataset (Hawco et al., 2016).

**Figure 4: Depth profiles of dissolved Co (navy circles) and labile dissolved Co (teal squares) along the GP15 Pacific transect. Suspected outliers are marked as an X.**





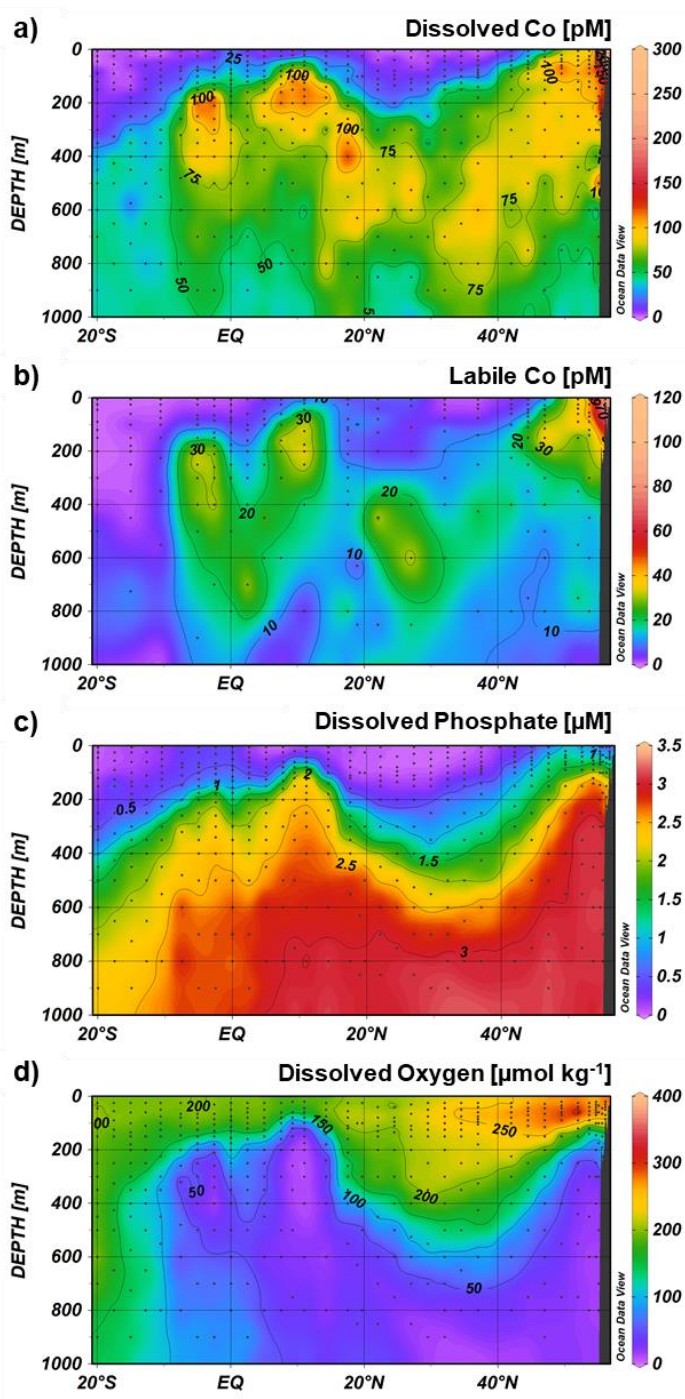

**Figure 5: Upper ocean transects of dissolved Co (a), labile dissolved Co (b), dissolved oxygen (c), and dissolved phosphate (d) along**
**the GP15 cruise track. Weighted-average gridding is used to interpolate between data points.**





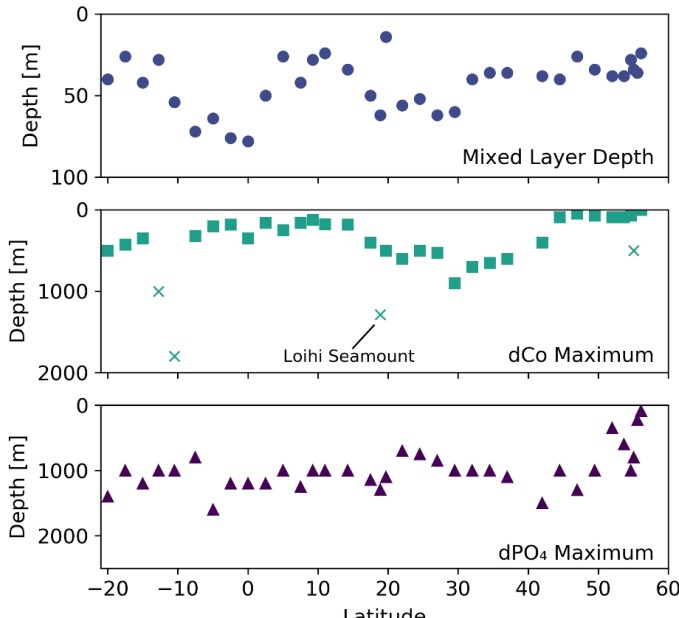

**Figure 6: Mixed layer depth, depth of the dCo maximum, and depth of the dPO4 maximum vs. latitude for the GP15 transect. Outlier dCo maxima depths are indicated with an 'x'.**

Dissolved Co was depleted in the surface Pacific Ocean (Fig. 5) due to phytoplankton uptake and utilization. The towfish samples, the surface-most samples collected for dCo around 2 m depth, typically represented a dCo minimum in the profile. Towfish dCo values were lowest in the North and South Pacific subtropical gyres, where average surface dCo reached near-zero values of $1.4 \pm 2$ pM (n = 10) and $0.7 \pm 2$ pM (n = 5), respectively. The surface dCo values were comparatively larger in the North Pacific subpolar gyre and equatorial Pacific, with average surface dCo values of $50 \pm 31$ pM (n = 5) and 8

$\pm 8$ pM (n = 10), respectively. The only exception to the observed dCo surface minimum was in the Alaskan coast; stations 1 and 2, the most coastal stations in the Gulf of Alaska, exhibited dCo maxima at the surface. The highest concentrations of dCo observed in the Pacific transect occurred in the towfish sample of station 1, which contained a dCo maximum of 576 pM (Fig. 4). This value is two orders of magnitude larger than the lowest observed surface values in the transect, highlighting the large and dynamic range of dCo within the Pacific. At this station, dCo was highest in the surface ocean and decreased with depth,

a profile typical of a scavenged-type element with a substantial coastal source in the surface ocean.

With the exception of the Alaskan coast and hydrothermally-influenced stations, dCo profile maxima were observed in the mesopelagic ocean due to biomass-associated Co remineralization and re-dissolution. Excluding the Alaskan coast surface maxima, dCo maxima of 54 to 167 pM were observed in intermediate depths between 45 and 900 m (Fig. 6). This wide range of depths of the dCo remineralization maxima is associated with regional variation of mixed layer depth and dCo

build-up within OMZs. Within the poorly ventilated OMZ regions, elevated dCo plumes are formed when there is a source of dCo via remineralization in the mesopelagic but little to no loss of dCo via scavenging onto Mn oxide particles (see Sect. 4.4). The average dCo profile maximum was shallowest in the water column in the upwelling regions of the North Pacific subpolar





gyre (76 ± 18 m, n = 6) and the equatorial Pacific (210 ± 74 m, n = 10), where the strongest influence from the North Pacific OMZ, ETNP, and ETSP were located. In these regions, the dCo maxima were shallower than the dPO$_4$ maxima that represents

the base of the nutricline, indicating that Co distribution within OMZs is somewhat decoupled from P remineralization, which is consistent with our understanding of elevated dCo in OMZs. In contrast, the average dCo profile maximum reached deeper depths in the North and South Pacific subtropical gyres (580 ± 150 m, n = 10 and 425 ± 75 m, n = 3, respectively) than in the regions affected by OMZs. These deeper mesopelagic dCo maxima follow a deepening nutricline in the subtropics, which is indicative of the downwelling environment, low biomass, and high light penetration of the gyres.

Scavenging of dCo within and below the mesopelagic ocean created low concentrations of dCo in the ocean interior. In the deep Pacific (depth ≥ 2000 m) the average dCo concentration was 27 ± 11 pM (n = 166), with large regions of the deep Pacific consistently under 25 pM dCo, particularly in the South Pacific. In contrast, the Atlantic Ocean has greater deep dCo concentrations on average (Dulaquais et al., 2014a; Noble et al., 2017; Wyatt et al., 2021), with the North and South Atlantic basins displaying deep dCo concentrations of 57 ± 12 pM (n = 166) and 40 ± 9 pM (n = 184), respectively (Schlitzer et al.,

2018). On GP15, dCo exhibited a heterogeneous distribution within the deep ocean, often with clear vertical regimes of elevated dCo that extend from the mesopelagic to the seafloor. For example, dCo is elevated at the equator (station 29), at 17.5º N off the coast of Hawaii (station 19), and in the North Pacific from 32º N–47º N (stations 14–8). These vertical regimes of elevated dCo are reminiscent of POC export dynamics, and likely represent the stabilization of deep dCo in a ligand-bound speciation. This vertical phenomenon is further explored in Sect. 4.6.

**3.3 Labile dissolved cobalt**

Labile dCo followed similar distribution patterns as dCo (Fig. 3b, 5b). Labile dCo was strongly depleted in the surface and deep interior Pacific Ocean, and was elevated in OMZs, coastal regions, the Loihi Seamount, and the mesopelagic, where remineralization occurs. The surface ocean south of the subpolar gyre contained exceptionally low concentrations of labile dCo, with average surface towfish samples (~2 m depth) of the North Pacific subtropical gyre and equatorial Pacific displaying

labile dCo values of 3 ± 5 pM (n = 6), and 1 ± 3 pM (n = 7), respectively. 14 of the 21 surface towfish samples analyzed for labile dCo contained concentrations below the detection limit of 2 pM, including all 3 of the South Pacific subtropical gyre towfish samples. The deep interior Pacific (≥ 2000 m depth) also contained very low to undetectable concentrations of labile dCo, with an average concentration of 2 ± 5 pM and 43 of the 64 deep samples analyzed containing labile dCo concentrations below the detection limit. Like dCo, labile dCo was elevated in the mesopelagic, particularly in the cores of the tropical Pacific

OMZ regions where labile dCo reached a maximum of 49 pM in the ETNP and 56 pM in the ETSP. Labile dCo was depleted in the North Pacific OMZ compared to total dCo, indicating that a large portion of the dCo plume within that low oxygen region was complexed by organic ligands.

The ratio of labile:total dCo in Pacific seawater was 0.33 ± 0.02 (R$^2$ = 0.55, n = 237; Fig. 7) when data points affiliated with coastal waters (station 1), OMZs ([O$_2$] < 50 µmol kg$^{-1}$), and hydrothermal activity (station 18.6 where depth > 1100 m)

were excluded because they exhibited unique labile:total dCo ratios. This "background" Pacific ratio is identical to the ratio





found in the South Pacific (0.33) on the GEOTRACES GP16 transect (Hawco et al., 2016). About two-thirds of dCo present in Pacific seawater existed in a strong ligand-bound speciation, likely as strong Co(III) complexes like vitamin $B_{12}$ and similar biologically-produced molecular species that are unable to exchange with strong ligands in experimental time frames. The other one-third of the dCo inventory existed as "free" or exchangeable, weakly-bound Co(II) species that are considered more

bioavailable and Mn oxide reactive (Baars and Croot, 2014; Saito et al., 2005). However, this 0.33 ratio was not consistent within all samples; of the 309 samples analyzed for labile dCo, 136 showed measurable total dCo but labile dCo values below the analytical limit of detection (2 pM), indicating near 100 % complexation. This caused the background seawater labile:total dCo slope to be depressed below the visual trend in higher labile and total dCo concentrations. Additionally, labile dCo values were found to be elevated compared to total dCo in the Loihi Seamount hydrothermal system and depressed with respect to

total dCo near the Alaskan coast, indicating the ratio of labile:total dCo varies among different dCo sources.

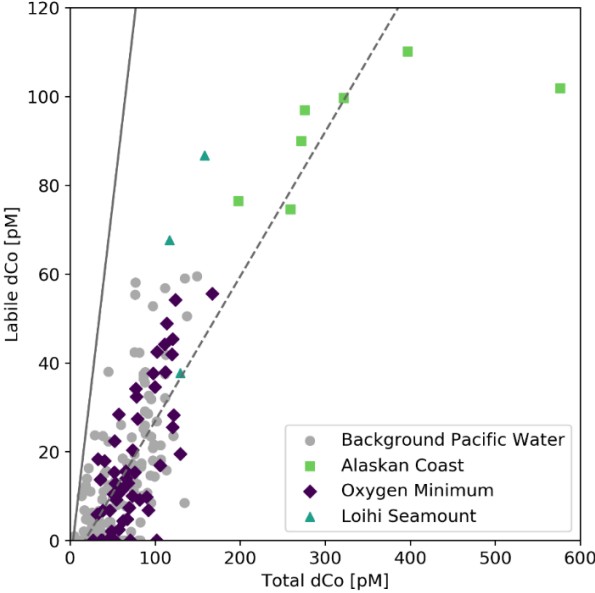

**Figure 7: Labile dCo vs. total dCo, with notable sources of dCo plotted in color, including station 1 along the Alaskan coast (green squares), oxygen minimum zones where O₂ concentrations are less than 50 μmol kg⁻¹ (purple diamonds), and samples from within the Loihi Seamount vent plume (teal triangles). A theoretical 1:1 line (solid) a linear regression of Background Pacific seawater**
**(dashed, slope = 0.33 ± 0.02, R² = 0.55, n = 237) are shown. Points to the right of the trendline represent greater dCo complexation compared to background seawater, and points to the left of the trendline represent more labile, un-complexed Co pools compared to background seawater.**

## 4 Discussion

### 4.1 The dCo and dPO₄ relationship depicts Co uptake, remineralization and scavenging

410        An analysis of dCo:dPO₄ ratios allows us to distinguish between the role of dCo as a nutrient-type and scavenged-type element. When dCo is correlated to dPO₄, its distribution is primarily controlled by biological uptake and remineralization as a micronutrient, and when dCo and dPO₄ are decoupled, its distribution is primarily controlled by additional processes





acting upon the dCo concentrations such as scavenging to oxide particles or an abiotic dCo source. These processes affecting dCo distributions can be depicted in dCo:dPO$_4$ vector space, where Co and PO$_4$ coupled processes like phytoplankton nutrient

uptake and remineralization result in vectors with positive slopes and opposite directionality, while decoupled processes like abiotic inputs and scavenging result in more vertical vectors that primarily show change in dCo concentration (Fig. 8a) (Noble et al., 2008; Saito et al., 2017). Co scavenging is also primarily a biologically-mediated process by oxidizing bacteria (Moffett and Ho, 1996), so biotic scavenging of dCo would in theory draw down a relatively small amount of dPO$_4$ via biological uptake. This is represented by a large negative dCo vector and a small negative dPO$_4$ vector, the sum of which is a scavenging

vector with a large positive slope drawn towards the lower left of the plot.

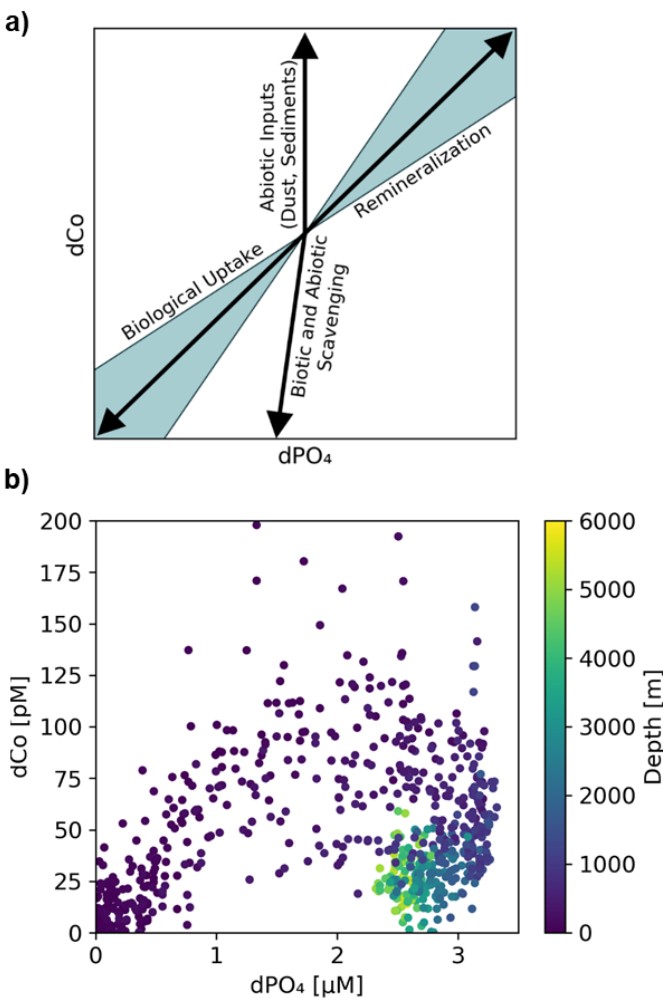

**Figure 8: (a) A vector schematic showing the relationship between dCo and dPO$_4$ concentrations and the effects of major oceanic processes on nutrient distribution. Biological uptake and remineralization can exhibit a range of stoichiometric relationships, depicted here by the blue shaded region. Adapted from Noble et al. 2008. (b) Observed dCo vs. dPO$_4$ along the GP15 transect.**



425          The dCo:dPO$_4$ relationship along the GP15 transect contained a relatively linear stoichiometry in the upper ocean and a "curl" towards the bottom right of the plot driven by mesopelagic and deep samples (Fig. 8b). The linear section in the upper ocean (0 to ~400 m) showed dCo distributions that were governed by phytoplankton uptake and remineralization, driving the coupling of dCo and dPO$_4$ ecological stoichiometry. The curl in the dCo:dPO$_4$ relationship occurred because of a shift in relative contributions of processes from remineralization to scavenging; the sum of the mesopelagic remineralization vector

(increasing dCo and dPO$_4$) and the scavenging vector (decreasing dCo, little change in dPO$_4$) resulted in a vector sum towards the lower right, creating a curl towards low dCo and high dPO$_4$ in DPW. The decoupling of dCo and dPO$_4$ due to interior scavenging is a characteristic feature of Co's hybrid geochemistry, and has been previously documented in the Pacific (Hawco et al., 2016), Atlantic (Noble et al., 2017; Saito et al., 2017), and Arctic Oceans (Bundy et al., 2020). To this date, the Southern Ocean is the only major basin that does not display a strong dCo scavenging vector (Noble et al., 2013; Oldham et al., 2021;

Saito et al., 2010). Below ~2500 m, the dCo:PO$_4$ relationship appeared to curl towards the origin. This phenomenon was driven by the mixing of water masses in the Pacific, namely the transition between the heavily remineralized NPDW, characterized by exceptionally high dPO$_4$, and the abyssal CDW, which contained relatively less dPO$_4$ even at greater depths in the Pacific (Fig. 3d). Thus, both dCo and dPO$_4$ were relatively depleted within the CDW mass compared to the intermediate NPDW mass, creating a deep Pacific "curl" towards the origin in the deep dCo:dPO$_4$ ratio. Note that mixing does not remove dCo or dPO$_4$

from the system; the CDW and NPDW masses simply contain different endmember dCo and dPO$_4$ concentrations.

         To further explore the dCo:dPO$_4$ relationship with depth, sample points along the Pacific transect were divided into 9 isopycnal bins by their potential density anomaly ($\sigma_0$), an analysis described in Hawco et al., 2018 (Fig. 9). Each bin contained sample points within a $\sigma_0$ range of 0.25 kg m$^{-3}$, with the exception of the least dense bin, which contained all samples with $\sigma_0$ < 26 kg m$^{-3}$. Stations 1 and 2, the Alaskan coastal stations, were removed from this analysis since they exhibited a clear surface

abiotic source vector and a negative upper ocean dCo:dPO$_4$ correlation (see Sect. 4.2), indicating the dCo:dPO$_4$ relationship was not primarily driven by uptake and remineralization. Dissolved Co and dPO$_4$ were positively correlated in the least dense bin ($\sigma_0$ < 26 kg m$^{-3}$, Fig. 9a), which represented the surface and upper ocean (depth range = 2–325 m, average depth = 84 ± 67 m). The dCo:dPO$_4$ slope was 82 ± 2 µM:M (n = 218, R$^2$ = 0.79), which is an estimate of the combined surface phytoplankton uptake stoichiometry and mesopelagic remineralization stoichiometry. This dCo:dPO$_4$ slope is comparable to the upper ocean

stoichiometry reported in the South Pacific (69 µM:M, R$^2$ = 0.89) (Hawco et al., 2016). In this study and the South Pacific study referenced, there appears to be one clear linear uptake and remineralization slope throughout the upper ocean depth range. This is distinct from studies in the Atlantic Ocean where the upper dCo:dPO$_4$ ratio varies with depth and dCo:dPO$_4$ slopes can be described as "accelerating" towards the surface ocean (Saito et al., 2017). High Atlantic dCo:PO$_4$ surface stoichiometry values have been reported up to 544 µM:M (depth = 40–136 m, R$^2$ = 0.79) (Saito et al., 2017) and 560 µM:M

(depth = 5 m, R$^2$ = 0.63) (Saito and Moffett, 2002), an order of magnitude higher than those reported in the Pacific and deeper Atlantic (27–67 µM:M, up to 200–900 m depth) (Dulaquais et al., 2014b; Noble et al., 2017; Saito et al., 2017). The extraordinarily high ecological stoichiometry in Atlantic surface samples has been attributed to strong draw-down of dPO$_4$ in the upper ocean coupled with increased use of Co in metalloenzymes, which is hypothesized to substitute for Zn as a cofactor



(Saito et al., 2017). This may be the case for alkaline phosphatase, an enzyme that cleaves phosphate groups from dissolved organic phosphorus molecules and is associated with P stress in phytoplankton (Cox and Saito, 2013; Quisel et al., 1996). Co has been shown to be the metallic cofactor for alkaline phosphatase in the hyperthermophile *Thermotoga maritima* (Wojciechowski et al., 2002), although Co substitution for Zn in marine cyanobacterial alkaline phosphatase is currently unconfirmed. The Pacific Ocean, in contrast, broadly shows less evidence for P limitation and more evidence for N and Fe stress in the oligotrophic gyres (Bonnet et al., 2008; Saito et al., 2014) and Fe and other trace metal stress in the upwelling

regions of the subpolar gyre and equatorial Pacific (Boyd et al., 2004; Coale et al., 1996; Moore et al., 2013; Tsuda et al., 2005; Ustick et al., 2021). This difference in extent of P stress in the Pacific basin compared to the Atlantic was likely the reason for the more linear relationship between dCo and dPO$_4$ throughout the upper Pacific Ocean.

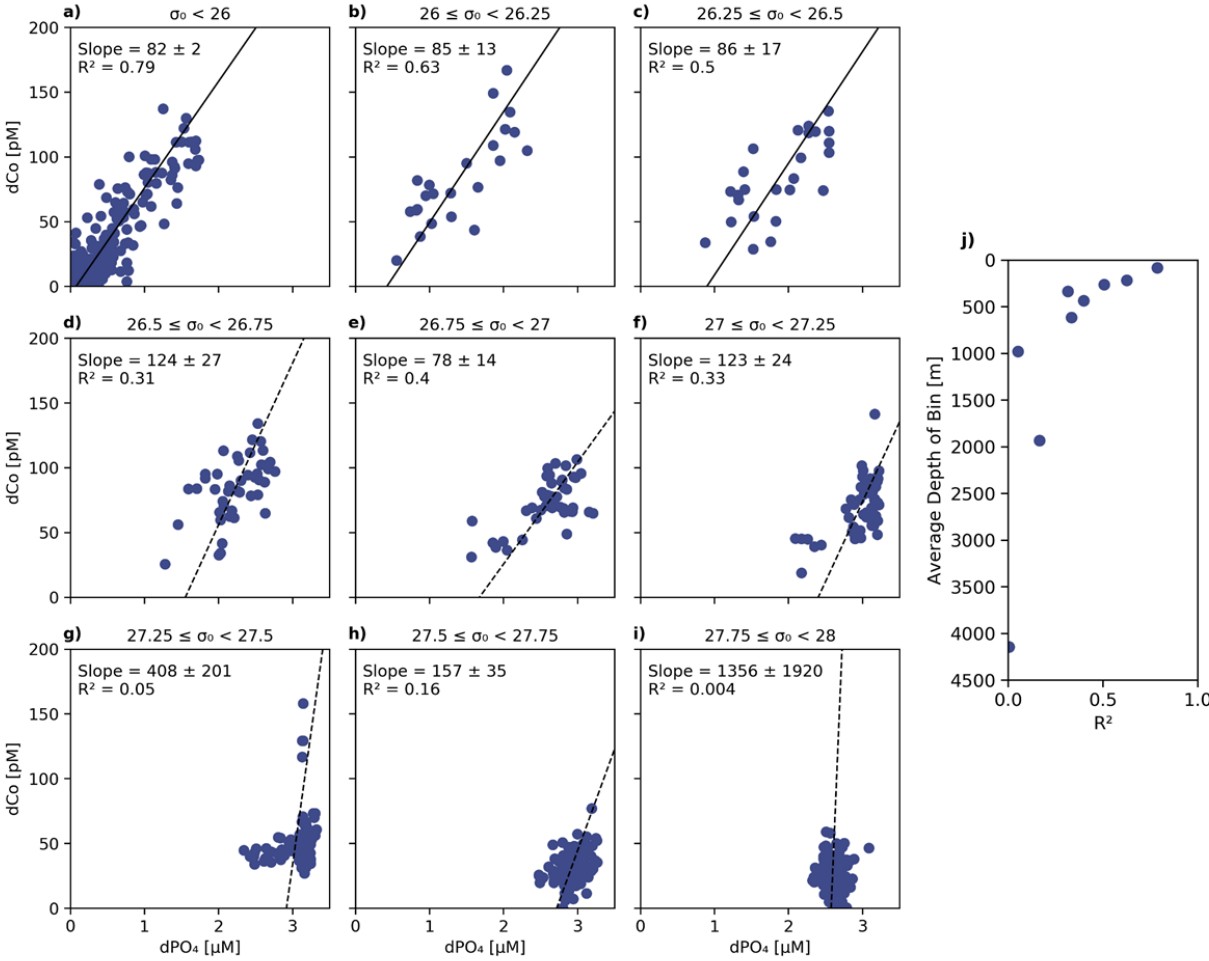

**Figure 9: (a-i) The dCo:dPO$_4$ relationship sorted into 9 isopycnal bins by potential density ($\sigma_0$) with a linear best-fit trendline. The**
**least dense samples (a) represent the surface ocean and phytoplankton dCo:dPO$_4$ nutrient ratio (82 ± 2 μM:M, n = 218, R$^2$ = 0.79).**
**The decreasing R$^2$ (j) in denser isopycnal bins represents the decoupling of dCo and dPO$_4$ at depth and increasing relevance of Co**
**scavenging. Samples with [dCo] > 200 pM and stations 1 and 2 have been omitted from this analysis.**





As the $\sigma_0$ bins increased in density, the correlations ($R^2$) between dCo and dPO$_4$ concentrations became less significant (Fig. 9j), indicating that dCo concentrations were becoming decoupled from dPO$_4$ in the deep Pacific due to Co scavenging.

In the second and third bins ($26 \leq \sigma_0 < 26.25$ and $26.25 \leq \sigma_0 < 26.5$ kg m$^{-3}$; Fig. 9b-c), the dCo:dPO$_4$ slope ($85 \pm 13$ and $86 \pm 17$ µM:M, respectively) is similar to that of the least dense bin ($82 \pm 2$ µM:M) but the slope error is increasing and the $R^2$ value is decreasing (0.63 and 0.50, respectively). Below the third density bin ($\sigma_0 \geq 26.5$ kg m$^{-3}$), the $R^2$ values of the correlations are extremely low and the slope values should not be interpreted as meaningful. The deep ocean density bins (Fig. 9g-i, average depths: 980–4144 m) displayed little dynamic range in dCo or dPO$_4$, indicating little to no significant remineralization or

scavenging signatures across the deep ocean despite the wide depth range represented by these bins. These findings are consistent with other studies that show Mn oxide formation and dCo scavenging primarily occurs along sharp density gradients in the mesopelagic and is not as significant in the ocean's deep interior where scavenging occurs at a slower rate (Hawco et al., 2018; Saito et al., 2017; Sunda and Huntsman, 1988).

## 4.2 The Alaskan coast as a source of ligand-bound dCo

In the coastal Gulf of Alaska at stations 1 and 2, the dCo and labile dCo profile maxima occurred at the ocean surface (~2 m depth; Fig. 10c). This dCo distribution is similar to that of the Arctic Ocean, where high surface dCo concentrations of a similar magnitude and a scavenged-type profile have been attributed to riverine inputs from the surrounding continents and sedimentary inputs from the Arctic shelf (Bundy et al., 2020). The station 1 towfish surface seawater sample contained the highest dCo and labile dCo measured in this study (576 pM and 102 pM respectively) and displayed a noticeably low

labile:total dCo ratio of 0.18, representing a source of predominantly ligand-bound, likely humic dCo to the surface water of the coastal Gulf of Alaska. In vector dCo:dPO$_4$ space (Fig. 8a), this riverine input is represented by an upward abiotic source vector. A strong dCo sedimentary source from a continental margin is consistent with a study along the Peruvian coast which found that dCo was positively correlated with $^{228}$Ra in surface waters, indicating a significant shelf-to-ocean source of dCo to the South Pacific (Sanial et al., 2018). Both dCo and labile dCo were negatively correlated to dPO$_4$ ($R^2 = 0.77$ and $R^2 = 0.57$,

respectively) and salinity ($R^2 = 0.81$ and $R^2 = 0.60$, respectively; Fig. 10a, b). The correlation to salinity is a strong indication that dCo from a coastal freshwater endmember is mixing with the pelagic marine dCo inventory. This relationship is consistent with studies that found that riverine ligand-bound dCo was linearly coupled with salinity in the Scheldt River estuary (Zhang et al., 1990), the Atlantic continental shelf off the Northeastern U.S. (Noble et al., 2017; Saito and Moffett, 2002), and the Amazon River output into the Atlantic (Dulaquais et al., 2014a). To estimate the dCo concentration of the riverine endmember,

the linear dCo:salinity relationship was extrapolated to a salinity of 0, resulting in a dCo concentration of $13.3 \pm 0.4$ nM and a labile dCo concentration of $1.69 \pm 0.09$ nM. This estimation assumes riverine dCo is mixing conservatively, which is unlikely to be true since other processes like scavenging and uptake are expected to occur at river mouths and coastal environments. The negative correlation with dPO$_4$ is striking since almost all other upper ocean samples are positively correlated with dPO$_4$ in the surface ocean due to nutrient uptake and remineralization (Sect. 4.1). Here, the negative dCo:dPO$_4$ correlation is likely





due to mixing as well; an endmember source of dCo at the surface with low dPO$_4$ is mixing with an open endmember that contains relatively lower dCo and higher dPO$_4$.

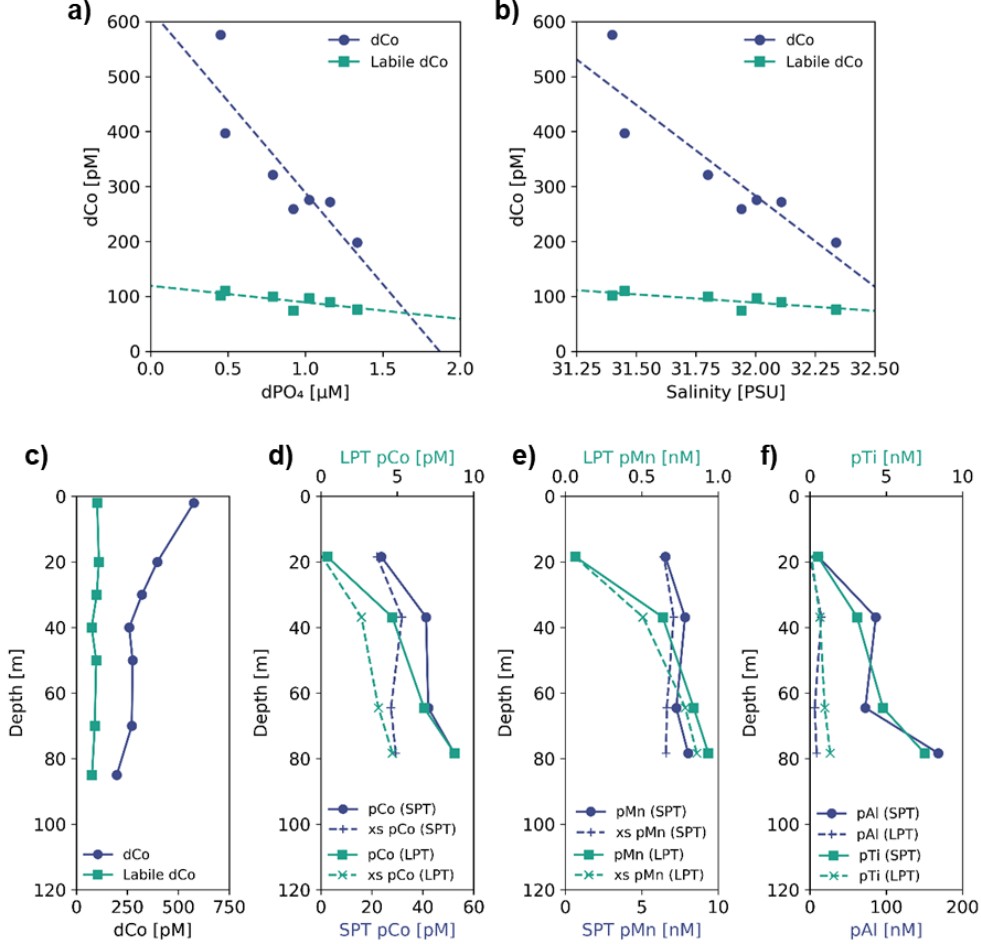

**Figure 10: Sample data and profiles from station 1, a coastal station in the Gulf of Alaska (56.06º N, 156.96º W). (a) dCo and labile dCo vs. dPO$_4$ show a significant negative correlation (slope = -430 ± 88 μM:M, R$^2$ = 0.77 and slope = -53 ± 18 μM:M, R$^2$ = 0.57,**
**respectively. (b) dCo and labile dCo vs. salinity show a significant negative correlation (slope = -406 ± 73 pM PSU$^{-1}$, R$^2$ = 0.81 and slope = -50 ± 15 pM PSU$^{-1}$, R$^2$ = 0.60, respectively). Depth profiles of (c) dCo and labile dCo, and of small particulate, total (SPT) and large particulate, total (LPT) (d) pCo and excess pCo (xs pCo), (e) pMn and excess pMn (xs pMn), and (f) pAl and pTi.**

Particulate Co, pMn, and the lithogenic tracers pAl and pTi were all elevated at depth, in stark contrast to the surface dCo maximum (Fig. 10d-f). The deep particulate metal maxima indicate the presence of a shelf-derived particulate flux that is
distinct from the dCo riverine surface flux. In the open ocean, dCo and pCo typically mirror each other in the upper water column (Noble et al., 2017), with biological uptake of Co resulting in low surface dCo and high surface pCo. The opposite distribution was observed at station 1, suggesting that dCo and pCo at this coastal station are more strongly influenced by external inputs rather than internal cycling, with higher surface dCo due to fresh riverine inputs and increasing pCo with depth due to sedimentary particle flux. Fe has been observed to behave similarly along the Alaskan coast, where terrestrial- and





glacial-derived dFe is transported by rivers to the surface coastal ocean (Crusius et al., 2011; Schroth et al., 2014), while the continental margins are a source of redox-sensitive pFe to the subsurface Gulf of Alaska (Lam et al., 2006; Lam and Bishop, 2008).

    The Alaskan coast thus acts as a source of both dCo and pCo to the coastal Gulf of Alaska, delivering a flux of Co to the Alaskan Stream, a westward-flowing coastal current along Southwest Alaska and the Aleutian Islands (Reed, 1984; Stabeno

et al., 1995). We observed little evidence that elevated Co distributions were advected off the shelf into the pelagic Gulf of Alaska; instead, much of the dCo inputs observed at stations 1 and 2 appear isolated to the coastline via transport within the Alaskan Stream, and elevated surface dCo was not observed at station 3 or beyond. This mechanism could work to isolate the HNLC region of the subpolar gyre from the metallic sources along the Alaskan coast and within the Alaskan Stream.

**4.3 Hydrothermalism and the Loihi Seamount**

Elevated dCo concentrations were observed at the Loihi Seamount (Station 18.6; 18.91º N, 155.26º W), with a dCo maximum of 158 pM and a labile dCo maximum of 87 pM at a depth of 1290 m. This maximum corresponded to a greater labile:total dCo ratio (0.55) compared to background Pacific Seawater (0.33; Fig. 7). An elevated dCo and labile dCo signature has previously been observed in the Hawaiian Kīlauea eruption of 2018 (Hawco et al., 2020), confirming previously observed dCo signals in the vicinity (Noble et al., 2008) and in the near-field plumes of the Juan de Fuca Ridge and the Atlantic Mid-

Ocean Ridge (Metz and Trefry, 2000; Noble et al., 2017).

    He isotopic samples were collected from the Ocean Data Foundation CTD (ODF) cast and not from the GEOTRACES CTD carousel (GTC) that the dCo samples were collected from. At the Loihi Seamount, the two CTD casts were completed at the sample station with latitudinal and longitudinal precision to within a thousandth of a decimal degree and were deployed within 1.5 hours of each other. However, the casts appear to be sampling different vent locations, indicated by anomalies in

the dFe and dMn data, which were collected from both casts. The GTC cast displayed trace metal (dCo, dFe, dMn) concentrations increasing towards the deepest sample at 1290 m, while trace metals (dFe, dMn) and xs$^3$He from the ODF cast displayed a shallower maximum around 1200–1250 m (Fig. 11a-d). This offset is likely due to a slight change in the wire angle of the CTD casts; even a small change in wire angle can cause a horizontal displacement of the CTD carousal of ~100 m. The Loihi Seamount sampling station was located near the center of Pele's Pit, a collapsed volcanic pit ~ 200 m deep located

towards the southern summit of the seamount. Pele's Pit contains several active hydrothermal vent fields within a radius of ~100 m of the Loihi station, and the casts likely sampled a combined hydrothermal signature from multiple vents (Clague et al., 2019; Jenkins et al., 2020). It is likely that the two CTD casts' wire angles and directions were offset in such a way that the GTC cast sampled a different signal of hydrothermal activity than the ODF cast.

    The offset between the GTC and ODF casts means that the dCo data and xs$^3$He data could not be directly compared

to show a hydrothermal-associated dCo source from the vent system. Instead, the dFe and dMn profiles collected from both casts were used to calculate a "Potential xs$^3$He" signal from the GTC cast; the dFe:xs$^3$He and dMn:xs$^3$He ratios from the ODF cast were calculated to be 7.6 ± 0.7 M:μM and 0.66 ± 0.06 M:μM, respectively (Fig. 11f). Samples from above 750 m were





excluded in order to isolate the hydrothermal relationships from unassociated upper ocean processes. Then, the GTC profile dFe and dMn values were multiplied by the dissolved metal:xs$^3$He ratio to find the potential xs$^3$He profile for the GTC cast

(Fig. 11e). This estimation assumed that the dFe:xs$^3$He and dMn:xs$^3$He ratios were consistent between all vents in Pele's Pit and thus are comparable between the two casts. This is a reasonable assumption to make for both Fe and Mn since both dissolved metals have been observed to be conservative in Loihi's hydrothermal plume (Jenkins et al., 2020), and all potential vent locations within the pit are likely sourced from the same sub-seafloor magma reservoir (Clague et al., 2019) and undergo similar low-temperature basalt leaching of trace metals.

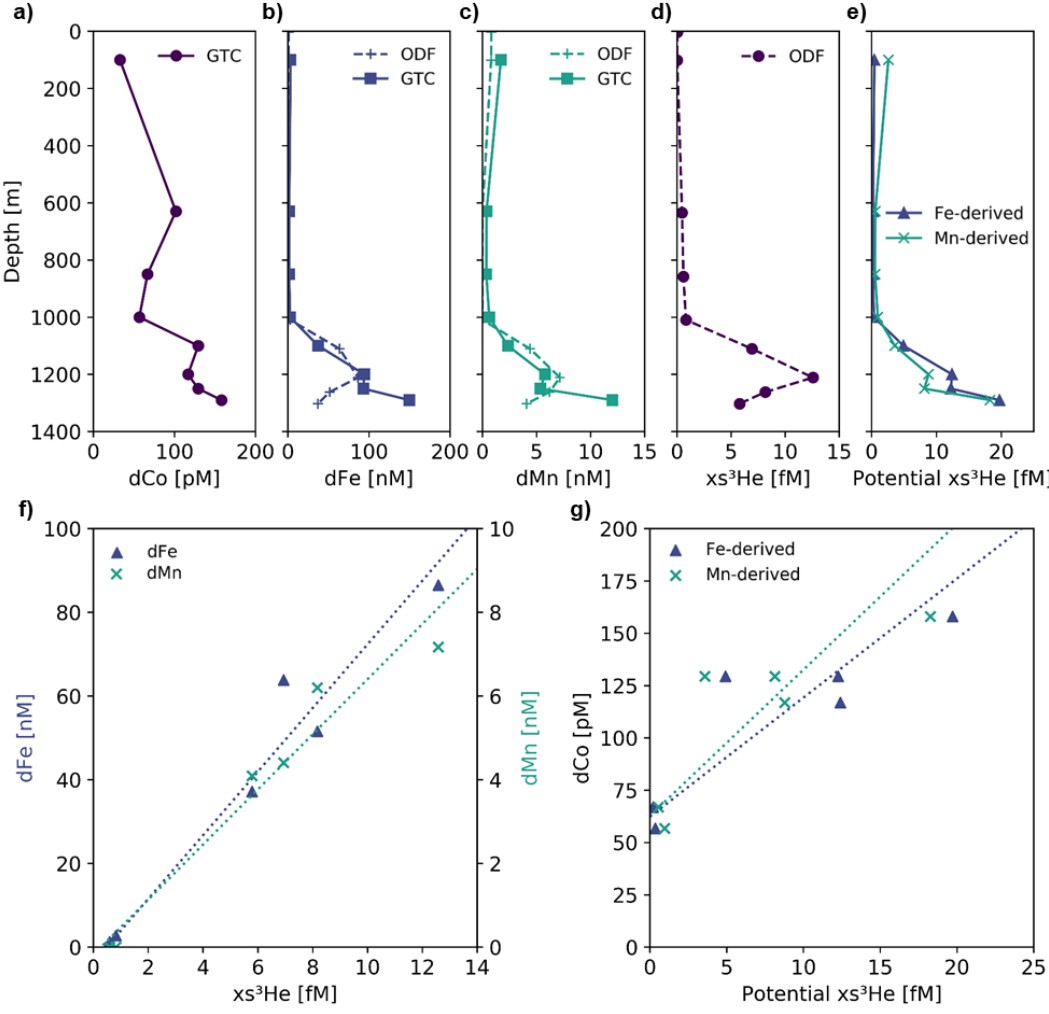


**Figure 11:** Profiles of (a) dCo, (b) dFe, (c) dMn, and (d) excess $^3$He (xs$^3$He) from the Loihi Seamount station (station 18.6) from both the trace-metal clean GEOTRACES CTD carousel (GTC, solid lines) and the general CTD from the Ocean Data Foundation (ODF, dashed lines). To estimate xs$^3$He values for the GTC cast, (f) the ODF cast's dFe:xs$^3$He and dMn:xs$^3$He ratios (7.6 ± 0.7 M:μM, $R^2$ = 0.95 and 0.66 ± 0.06 M:μM, $R^2$ = 0.95 respectively) were used to predict (e) potential xs$^3$He values with respect to the GTC dFe and

dMn values. (g) The GTC dCo:potential xs$^3$He ratios were 6 ± 1 M:mM (Fe-derived, $R^2$ = 0.78) and 7 ± 2 M:mM (Mn-derived, $R^2$ = 0.71). For the trace metal:xs$^3$He regressions, only samples below 750 m were used to isolate the hydrothermal signal from surface processes.





Dissolved Co was positively correlated with the estimated $^3$He anomaly at the Loihi Seamount (Fig. 11g). The Fe-derived and Mn-derived GTC dCo:potential xs$^3$He ratios were found to be similar to each other, with values of  $6 \pm 1$ M:mM
($R^2 = 0.78$) from the Fe-derived estimate and $7 \pm 2$ M:mM ($R^2 = 0.71$) from the Mn-derived estimate. This correlation shows that there is a local dCo source from the Loihi Seamount that is clearly associated with hydrothermal activity, and that the local dCo source is exported above the sill height of Pele's Pit (~1100 m) to the surrounding water column. The $^3$He flux from the Loihi Seamount vent was estimated to be $10.4 \pm 4$ mol year$^{-1}$ during the GP15 expedition (Jenkins et al., 2020), suggesting a local Co flux of $6 \pm 3 \times 10^4$ mol Co year$^{-1}$ using the Fe-derived estimate, or $7 \pm 3 \times 10^4$ mol Co year$^{-1}$ using Mn-derived
estimate. This local dCo source is one of the few hydrothermal Co flux observations that shows transport of dCo from an active vent without directly sampling hot vent fluids. The estimated Co flux would represent 3 to 4 % of the global hydrothermal Co flux, which has been estimated to be ~$2 \times 10^6$ mol Co year$^{-1}$ (Swanner et al., 2014). This percentage is surprisingly high for one hydrothermal vent system, but it should be noted that the $^3$He flux at the Loihi Seamount was also estimated to be ~2 % of the global deep hydrothermal flux (Jenkins et al., 2020), and that the majority of the hydrothermal dCo source is expected
to be quickly scavenged.

The Loihi Seamount was the source of a distal plume of elevated xs$^3$He that stretched across an estimated $1 \times 10^{12}$ m$^2$ area of the Pacific basin (Jenkins et al., 2020). Elevated dCo concentrations associated with the plume were only found at one adjacent station; station 19 displayed a dCo maximum of 67 pM at 1145 m depth, 380 km away from the Loihi Seamount. In this sample, labile dCo was found to be below the analytical detection limit of 2 pM, indicating that all dCo in the sample was
ligand-bound. This finding is consistent with the hypothesis that labile dCo is quickly scavenged or bound by ligands before it is able to be transported far via the hydrothermal plume. However, unlike xs$^3$He, dMn and dFe, concentrations of dCo were not found to be conservative within the distal plume. When the concentrations of dCo and xs$^3$He from distal stations (stations 14–25) were interpolated to the core plume depth of 1100 m, they were not found to be negatively correlated ($R^2 = 0.28$), which would have been an indicator of dCo transport throughout the plume and possible upwelling to the surface ocean. Thus,
although ligand-bound dCo showed some evidence of travel within the hydrothermal plume, dCo does not act conservatively within the plume, likely due to local scavenging before it is transported far from the hydrothermal source.

Hydrothermal vents are notoriously heterogenetic, and the variations in vent fluid metal compositions between individual vents, ocean basins, and over time contribute to a relatively unconstrained dCo hydrothermal flux. Hydrothermalism has been considered a negligible source of Co to the marine system since labile dCo is particularly susceptible to scavenging.
It has been estimated that hydrothermal activity contributes $1.2 \times 10^{11}$ g Co kyr$^{-1}$ to the marine Co cycle, only ~2.4 % of the total marine Co budget (Swanner et al., 2014). A more recent study to further constrain the global marine Co cycle adjusted this hydrothermal estimate to be only ~0.3 % of the total marine Co flux when other dCo sources were added or increased (Hawco et al., 2018). Here, we observed local vertical transport of dCo to the surrounding water column, but not horizontal transport of dCo to the greater Pacific basin. Our findings corroborate the evidence that dCo from Pacific hydrothermal inputs
is likely relatively local and minor, but also suggests that ligand-bound dCo might travel farther within the hydrothermal plume than previously expected.



### 4.4 Elevated dCo in oxygen minimum zones

The phenomenon of elevated dCo in $O_2$ depleted regions has been documented throughout the global oceans, including in the North Atlantic, South Atlantic, and South Pacific tropical OMZs (Baars and Croot, 2014; Hawco et al., 2016;

Noble et al., 2012, 2017). These hotspots are created when there is a source of dCo to the mesopelagic via remineralization of sinking phytoplankton biomass and/or sedimentary flux processes but a suppressed sink of dCo via scavenging onto Mn oxides particles (Hawco et al., 2016). Manganese oxides are slow to form due to the low $O_2$ conditions, and the reductive dissolution of advected manganese oxide particles is favored (Johnson et al., 1996; Sundby et al., 1986). Thus, dCo is able to build up in low $O_2$ regions within the mesopelagic and basin interior, creating plumes of dCo associated with OMZs (Hawco et al., 2016).

The dCo:$O_2$ correlation observed in and around OMZs represents both this accumulation of dCo in low oxygen waters as well as its advection and mixing into surrounding oxic waters.

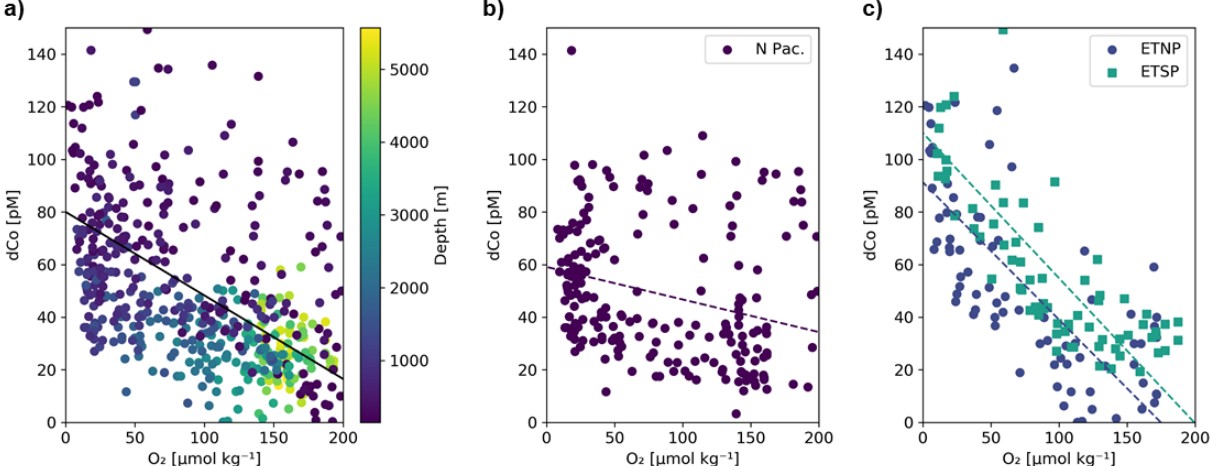

**Figure 12: (a) Dissolved vs. $O_2$ relationships within intermediate and deep GP15 samples. Datapoints from the surface (< 150 m) and that contain > 200 µmol kg$^{-1}$ $O_2$ have been ommited. The solid black line represents the integrated dCo:$O_2$ correlation across the**
**transect (slope = -0.32 ± 0.03 µM:mol kg$^{-1}$, n = 493, $R^2$ = 0.22). dCo:$O_2$ correlations are also shown by OMZ region, including (b) the North Pacific OMZ (stations 3–18), (c) the ETNP (stations 20–25) and ETSP (stations 29–34). The ETNP and ETSP displayed significant negative dCo:$O_2$ slopes of -0.52 ± 0.06 µM:mol kg$^{-1}$ (n = 77, $R^2$ = 0.51) and -0.55 ± 0.05 µM:mol kg$^{-1}$ (n = 71, $R^2$ = 0.59), respectively. The North Pacific displayed no clear linear correlation between dCo and $O_2$ (n = 205, $R^2$ = 0.05).**

Dissolved Co and $O_2$ showed a weak negative correlation throughout the transect below 150 m depth and 200 µmol
kg$^{-1}$ $O_2$, and this relationship became more significant within the ETNP (stations 20-25; $R^2$ = 0.51) and ETSP (stations 29-34; $R^2$ = 0.59; Fig. 12). The dCo:$O_2$ slopes within the ETNP (-0.52 ± 0.06 µM:mol kg$^{-1}$) and ETSP (-0.55 ± 0.05 µM:mol kg$^{-1}$) reflect the build-up of dCo in OMZs due to heterotrophic remineralization in the mesopelagic, suppressed scavenging in low $O_2$ environments, and mixing between OMZ waters and surrounding oxic waters. These dCo:$O_2$ slopes are very similar to those observed in the North and South Atlantic, but slightly higher than observed in the ETSP along the GP16 expedition

(Table 2) (Baars and Croot, 2014; Hawco et al., 2016; Noble et al., 2012, 2017). However, the dCo:$O_2$ values observed along the GP15 transect cannot necessarily be extrapolated to the entire ETSP or ETNP because the transect only captures the tail

 

ends of the equatorial OMZs where the O₂ depletion and subsequent dCo enrichment are not as strong as they are further east
along the margin of the Americas (Hawco et al., 2016). This consideration may explain the relatively low $R^2$ values of the
GP15 correlations compared to prior zonal transects, although the persistence of the OMZ systems > 5000 km from its core
along the coast demonstrates the strength and geochemical influence of the tropical Pacific OMZs.

**Table 2: Dissolved Co vs. O₂ correlation within OMZs in the Pacific GP15 transect (this study) and others.**

| Oceanographic Location | Depth Range [m] | n | dCo:O₂ [µM:mol kg⁻¹] | $R^2$ | Reference |
|---|---|---|---|---|---|
| Integrated Pacific Ocean[a] | ≥ 150 | 493 | -0.32 ± 0.03 | 0.22 | This Study |
| Tropical North Pacific (ETNP) | ≥ 150 | 77 | -0.52 ± 0.06 | 0.51 | This Study |
| Tropical South Pacific (ETSP) | ≥ 150 | 71 | -0.55 ± 0.05 | 0.59 | This Study |
| North Atlantic | 300 - 800 | 73 | -0.56 | 0.89 | Noble et al. 2017 |
| South Atlantic Angola Dome | 300 - 800 | n/a | -0.56 | 0.73 | Noble et al. 2012 |
| Tropical North Atlantic | 0 - 800 | n/a | -0.39 | 0.88 | Baars and Croot 2015 |
| Tropical South Pacific (ETSP) | 200 - 5500 | n/a | -0.34[b] | 0.75 | Hawco et al. 2016 |

[a] Integrated Pacific Ocean refers to all samples along the GP15 transect ≥ 150 m depth and ≤ 200 µmol kg⁻¹ O₂.
[b] The dCo:O₂ correlation was originally reported as -0.33 µM:M, and was adapted to units of µM:mol kg⁻¹ using a conversion of
1.025 kg L⁻¹.

In contrast, the North Pacific OMZ showed little to no correlation between dCo and O₂ ($R^2$ = 0.05), which likely
depressed the whole-transect dCo:O₂ correlation. This lack of correlation was unexpected given the robust relationship of dCo
and O₂ in OMZs observed throughout the world. These results speak to a unique mechanism governing dCo distributions in
the relatively stagnant Pacific OMZ compared to the equatorial OMZs, which could be driven by their differences in (1) anoxic
intensity or (2) Co margin sources.

Compared to the North Pacific OMZ, the ETNP and ETSP are more intense oxygen minimum regions closer to the
continental margin, where O₂ reaches concentrations that are < 1.5 µmol kg⁻¹ (Karstensen et al., 2008). The North Pacific does
not reach the same strong low O₂ conditions as the coastal equatorial OMZs; the lowest O₂ value measured in the subtropical
North Pacific on this expedition was 8 µmol kg⁻¹. The North Pacific OMZ would therefore support a greater kinetic rate of
Mn-oxide formation than the more anoxic ETNP and ETSP (Johnson et al., 1996), and thus would exhibit a greater sink of
dCo via scavenging. However, the presence of a deep, ligand-bound dCo signal in the North Pacific suggests that some dCo
scavenging is depressed within the oxygen minimum, indicating the importance of dCo complexation with organic ligands in
this region (see Sect. 4.6). Additionally, the Atlantic equatorial OMZs reach similar oxygen minimums as those observed in
the North Pacific OMZ and still exhibit negative correlations between dCo and O₂ (Noble et al., 2012, 2017), so the difference
in anoxic intensity cannot solely account for the lack of correlation in the North Pacific OMZ.

The North Pacific OMZ also lacks a single point-source of dCo and pCo within its low O₂ waters, compared to the
coastal Co sources from the eastern boundary margins that are advected westward within the equatorial OMZs. In the ETNP
and ETSP, coastal dCo inputs are protected from scavenging and margin pCo inputs are available for reduction and
remobilization within the low O₂ concentrations of the OMZ as this elevated dCo signal is advected from the coast to the



pelagic ocean (Hawco et al., 2016). The North Pacific, in contrast, has a number of diffuse dCo sources from the surrounding
continental margin and advection from the Kuroshio Extension current (Zheng et al., 2019), but there is little evidence that
coastal sources are significantly tranported to the pelagic North Pacific (Sect. 4.2). Additionally, the OMZ does not extend to
the continental margin, and so any advected dCo source will be more vulnerable to loss via remineralization and scavenging
as it travels between its coastal origin and the North Pacific OMZ. This lack of an intersection between the OMZ and a
continental margin is likely a key reason why we do not observe a correlation between dCo and $O_2$ in the North Pacific.

**4.5 Implications of deoxygenation on the Pacific dCo inventory**

The extent and intensity of Pacific OMZs have been increasing over the past 50 years due to global ocean warming,
and OMZs are predicted to continue to expand throughout the next century (Matear and Hirst, 2003; Stramma et al., 2008).
The waters surrounding OMZs are predicted to have the greatest $O_2$ loss rate; over a 50-year period, Stramma et al. 2008
estimated an $O_2$ loss trend ($\Delta O_2/\Delta t$) of -0.13 ± 0.32 µmol kg$^{-1}$ yr$^{-1}$ in the core of the equatorial Pacific OMZs (5º N to 5º S,
105º W to 115º W), compared to a more negative $\Delta O_2/\Delta t$ of -0.19 ± 0.20 µmol kg$^{-1}$ yr$^{-1}$ farther west on the outskirts of the
OMZs (5º N to 5º S, 165º W to 175º W). Note that the $\Delta O_2/\Delta t$ estimates have a high error, which makes the prediction of future
effects due to deoxygenation uncertain. The GP15 expedition passed between the locations of these two equatorial Pacific
estimates, and it is reasonable to assume $\Delta O_2/\Delta t$ within the equatorial OMZs at the longitude of the GP15 transect (152º W)
falls somewhere between the two estimates.

Deoxygenation is expected to affect redox-sensitive trace element cycles in and around expanded OMZs. Compared
to other trace metal cycles, the Co cycle may be notably affected by decreasing $O_2$ concentrations because of its exceptionally
small dissolved inventory; expanding OMZs are predicted to increase the dCo inventory and decrease the Co scavenging flux,
representing a significant perturbation to the marine Co cycle. In the next 100 years, the dCo inventory is estimated to increase
by ~10 % in the South Atlantic OMZ (Noble et al., 2012), and by ~20 % in the North Atlantic OMZ (Noble et al., 2017).

Using the $O_2$ loss trends from Stramma et al. 2008 and the dCo:$O_2$ linear trends calculated above, we estimated the
increase to the dCo inventory within the 152º W cross-sectional transect of the ETNP and ETSP over the next 100 years (Table
3). Assuming the linear relationship observed between dCo:$O_2$ will be constant over the next century, the dCo inventories
within the equatorial Pacific OMZs are estimated to increase at a rate ($\Delta dCo/\Delta t$) of 0.068 to 0.11 pM yr$^{-1}$. To predict the effect
of this rate of dCo change in the Pacific, the upper ocean (≥ 1000 m) dCo inventory was estimated over the stations within the
ETNP and ETSP via trapezoidal integration of dCo station profiles over depth and latitude. The estimated dCo inventories
were determined within upper ocean boxes of the Pacific with a volumetric width of 1 m$^3$, a depth of 1000 m, and a longitude
of 152º W. The GP15 cross section of the ETNP has a present dCo inventory ($I_{present}$) of 52 mol and is predicted to increase by
13–19 % over the next 100 years, and the cross section of the ETSP has an $I_{present}$ of 38 mol dCo and is predicted to increase
by 19–28 %.





**Table 3: Predicted effects of deoxygenation on the dCo inventories (I) of the upper 1000 m of the Eastern Tropical North Pacific (ETNP) and Eastern Tropical South Pacific (ETSP) oxygen minimum zones (OMZs) along the GP15 transect at 152° W. For**
**comparison, ΔdCo/Δt was estimated using the dCo:$O_2$ relationship from the GP16 expedition (Hawco et al., 2016). $I_{present}$ represents the current dCo inventory integrated over the OMZ stations, and $I_{100\ yr}$ represents the estimated change in the dCo inventory in 100 years. $V_{section}$ represents the volume of each GP15 section sampled, with a width of 1 m and a depth of 1000 m.**

| OMZ | Stations Included | Latitudinal Range | $V_{section}$ [m$^3$] | dCo:$O_2$ [μM:mol kg$^{-1}$] | $\Delta O_2/\Delta t^a$ [μmol kg$^{-1}$ yr$^{-1}$] | ΔdCo/Δt [pM yr$^{-1}$] | $I_{present}$ [mol] | $I_{100\ yr}$ [mol] | I % Increase |
|---|---|---|---|---|---|---|---|---|---|
| ETNP | 20, 21, 22, 23, 25 | 14.25° to 5° N | $10 \times 10^8$ | -0.52 ± 0.06 | -0.13 ± 0.32 | 0.068 ± 0.17 | 52 | 58 ± 17 | 13% |
| | | | | | -0.19 ± 0.20 | 0.10 ± 0.11 | | 62 ± 10 | 19% |
| ETSP | 29, 31, 33, 34 | 0° N to 7.5° S | $8.3 \times 10^8$ | -0.55 ± 0.05 | -0.13 ± 0.32 | 0.072 ± 0.18 | 38 | 45 ± 18 | 19% |
| | | | | | -0.19 ± 0.20 | 0.11 ± 0.11 | | 48 ± 11 | 28% |
| GP16 ETSP | All | | | -0.34 | -0.13 ± 0.32 | 0.044 ± 0.11 | | | |
| | | | | | -0.19 ± 0.20 | 0.064 ± 0.068 | | | |

**$^a$ Pacific deoxygenation trends (ΔO$_2$/Δt) are from Stramma et al. 2008, Table 1. ΔO$_2$/Δt rates were estimated within the core of the Pacific equatorial OMZs (-0.13 ± 0.32 μmol kg$^{-1}$ yr$^{-1}$; 5° N to 5° S, 105° W to 115° W) and along their western boundary (-0.19 ± 0.20**
**μmol kg$^{-1}$ yr$^{-1}$; 5° N to 5° S, 165° W to 175° W).**

This increase in the dCo inventory is similar or slightly higher than those predicted in the Atlantic, likely because the more pelagic regions of the OMZ captured by the GP15 transect are predicted to experience higher rates of deoxygenation than those closer to the coastal core of the equatorial OMZs. Similarly, the calculated ΔdCo/Δt of the ETSP on the GP15 expedition is greater than is predicted when we use the dCo:$O_2$ relationship from the GP16 expedition (-0.34 μmol kg$^{-1}$ yr$^{-1}$),
which estimates a ΔdCo/Δt of 0.044 to 0.064 pM yr$^{-1}$ using the same ΔO$_2$/Δt values; the GP16 expedition sampled closer to the South American coastline and the core of the ETSP than the GP15, and so a less dramatic dCo increase trend may be more representative of the near-shore equatorial OMZs. Estimating the open-ocean OMZ trend appears particularly important, as the low dCo inventory of the more pelagic South Pacific is acutely sensitive to deoxygenation, where the ETSP expansion could increase the dCo inventory at 152° W by as much as 28 % over the next 100 years.

We recognize these are simple back-of-the-envelope calculations, but they nevertheless may be useful to inform the direction, if not the exact magnitude, of future changes in dCo inventory. The estimates presented here all have a high error associated with their calculation, which stems from the uncertainty of the original ΔO$_2$/Δt rates from Stramma et al. 2008. The rates of Pacific deoxygenation and ocean warming are relatively unconstrained, making it difficult to predict the effects of global climate change on the Pacific Co cycle. Furthermore, the assumption that the dCo:$O_2$ relationship will stay constant
over the next 100 years of global change is not necessarily valid; excess dCo builds up in low oxygen waters, and much of the observed dCo:$O_2$ correlation can be considered a mixing line between the OMZ source and oxic waters. The dCo source of the equatorial OMZs is primarily unscavenged dCo advected from continental shelf margins, which is driven by the surface area of the shelf sediments exposed to low $O_2$ waters (Hawco et al., 2016). As the equatorial OMZs expand, this OMZ-shelf surface area will also expand, likely increasing the dCo source to the ETNP and ETSP. Thus, the dCo:$O_2$ slope presented here could
become more negative over time if the OMZ (dCo source) endmember increased but the oxic (low dCo) endmember stayed





similar or constant, leading to greater increases of the dCo Pacific inventory than estimated here. We recommend future modeling, experimental, and observational work to better constrain the effect of deoxygenation on the oceanic Co inventories.

Although we did not observe a significant $dCo:O_2$ relationship in the North Pacific OMZ (see Sect. 4.4), Stramma et al. 2008 estimates that the North Pacific has had a higher rate of $O_2$ loss over the past 50 years than all other regions studied (-

0.39 to -0.70 µmol $kg^{-1}$ $yr^{-1}$). Strong North Pacific deoxygenation trends may drive relatively large increases in the dCo inventory over the next century as $O_2$ concentrations decrease and the region becomes a source of dCo via Mn-oxide reduction. Currently, the potential dCo trend in the North Pacific is difficult to estimate, but future deoxygenation could result in greater increases in the dCo inventory here than any other ocean basin. More research on the North Pacific OMZ and deoxygenation is required to determine the extent of future perturbation to the North Pacific Co cycle.

**4.6 The presence of deep, stable dCo in the North Pacific**

One of the most unexpected features in the GP15 dCo transect was the discovery of stable, ligand-bound dCo in the deep interior of the Pacific Ocean, particularly in the North Pacific basin. These higher dCo concentrations do not appear to be due to analytical error or blank correction error, and instead appear to be robust evidence for the presence of complexed dCo below the mesopelagic that has been protected against scavenging. This distribution of dCo is unusual below ~2000 m

because dCo tends to be removed via scavenging throughout the mesopelagic and deep ocean. Strong organic ligands present in the deep ocean have also been observed for copper (Cu) in the North Pacific subarctic gyre where ligand-bound Cu dominated the dissolved Cu speciation to at least 3000 m depth (Moffett and Dupont, 2007).

The discovery of elevated dCo in the North Pacific was unexpected, since dCo has been shown to be scavenged along deep thermohaline circulation with deep dCo concentrations decreasing with [14]C age (Hawco et al., 2018). When deep GP15

samples (> 3000 m) were paired with their nearest radiocarbon age measurement from the GLODAP database and compared to data from previous GEOTRACES expeditions (Fig. 13a-b; Table 4), many GP15 samples from the North Pacific appeared to deviate from the expected trend of decreasing dCo with water mass age. Deep samples along this transect are estimated to have a conventional radiocarbon age of 918–1645 years, with the intermediate North Pacific containing the oldest waters (>1600 years). The low ventilation of these deep waters can also be seen in the average AOU value at each station below 1000

m depth (Fig. 13e), which displays a steady increase of oxygen utilization with latitude along the transect. Thus, intermediate and deep North Pacific waters were expected to have the lowest, most depleted concentrations of dCo as they are the some of the oldest, least ventilated ocean waters in the world and have had a relatively long timeframe to scavenge their available dCo. While many of the deep GP15 samples, particularly in the South and equatorial Pacific, fell along this predicted trend of slow but steady scavenging within aging water masses, many samples from the North Pacific were relatively high in dCo despite

their age. This finding suggests a source of dCo to the deep North Pacific, superimposing a regional source process on top of a global circulation scavenging process.





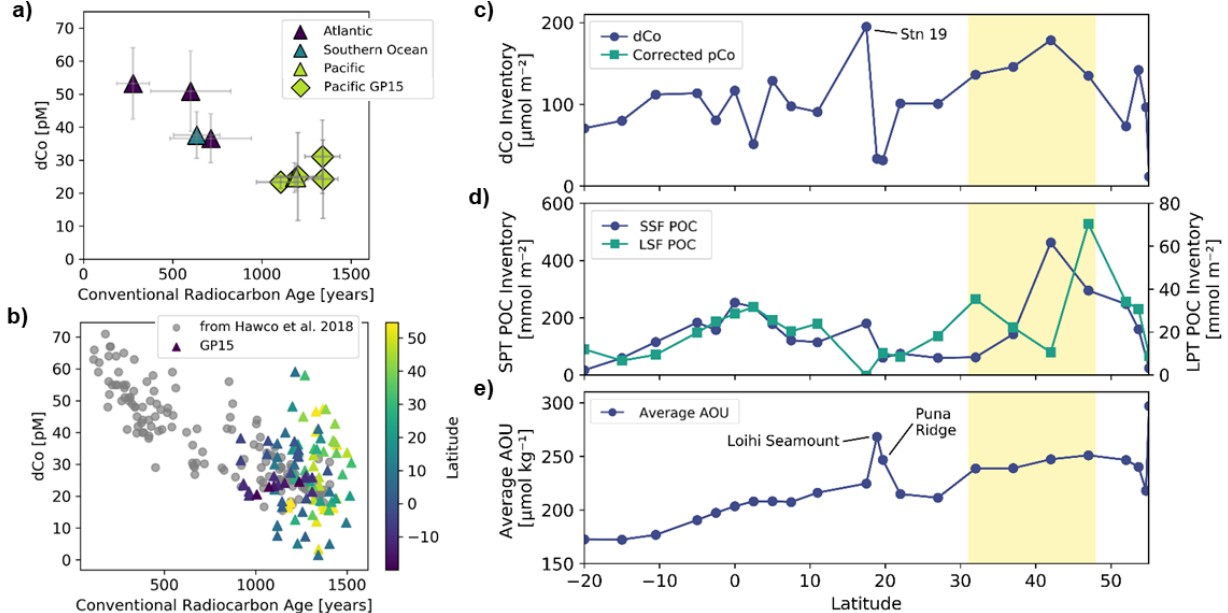

**Figure 13: (a) Average dCo concentrations vs. radiocarbon age of the deep ocean (≥ 3000 m) by oceanographic basin with error bars of standard deviation. Data from GEOTRACES expeiditions other than GP15 (triangles) are from Hawco et al. 2018 Table 2. (b) dCo vs. radiocarbon age of individual GEOTRACES dCo samples matched with their nearest GLODAP radiocarbon measurement. (c-d) Deep ocean (≥ 1000 m) inventories of dCo, small particulate, total (SPT) particulate organic carbon (POC) and large particulate, total (LPT) POC for each deep station profile. Station inventories were estimated by trapezoidal integration below 1000 m depth. (e) Average deep (≥ 1000 m) apparent oxygen utilization (AOU) at each station. The yellow boxes highlight the region in the North Pacific where the deep dCo inventory is elevated.**

Little excess pCo or pMn was observed in the North Pacific, suggesting few Mn-oxide particles were present in this region (Fig. 14). As noted in Sect. 4.2, pCo and dCo phases often exhibit mirror distributions; where there is high dCo, there tends to be low pCo and vice versa. This finding is consistent with a low rate of Mn and Co scavenging in the mesopelagic and deep North Pacific. Additionally, POC concentrations were elevated at depth in the North Pacific subpolar gyre, and the elevated ligand-bound dCo observed in this region partially overlaps with this deep POC signal, although the deep dCo signal extends farther south than the deep POC signal. Therefore, the unexpected deep dCo signal may be associated with regional POC export.

To better examine deep ocean trends with latitude, the dCo and POC station inventories were estimated using trapezoidal integration below 1000 m. The inventory estimations show consistently elevated inventories of dCo in the North Pacific (Fig. 13c). In the same region, there was dramatically elevated particulate organic carbon (POC) inventories at some stations. The SPT POC inventory spiked to 463 mmol m$^{-2}$ at station 10 (42º N, 152º W), while the LPT POC inventory had a similar maximum of 70 mmol m$^{-2}$ a little farther north at station 8 (47º N, 152º W; Fig. 13d).





**Table 4: Average conventional radiocarbon age, apparent oxygen utilization (AOU), and dCo of the deep Pacific Ocean (> 3000 m**
**depth). GP15 samples were matched with their nearest $\Delta^{14}C$ and AOU observations from the GLODAP database.**

| Region | n | Average Radiocarbon Age [Years] | Average AOU [$\mu$mol kg$^{-1}$] | Average dCo [pM] |
|---|---|---|---|---|
| N Pac. Subpolar Gyre | 20 | $1342 \pm 85$ | $196 \pm 10$ | $24 \pm 12$ |
| N Pac. Subtropical Gyre | 29 | $1340 \pm 98$ | $187 \pm 12$ | $31 \pm 11$ |
| Equatorial Pac. | 32 | $1200 \pm 130$ | $174 \pm 14$ | $25 \pm 13$ |
| S Pac. Subtropical Gyre | 14 | $1110 \pm 140$ | $159 \pm 11$ | $23 \pm 2$ |

One compelling explanation for the complexed deep dCo presence is the wide North Pacific OMZ. Unlike the shallower equatorial OMZs of the Atlantic and Pacific, this low oxygen zone stretches across the majority of the North Pacific
mesopelagic. Although the dCo:$O_2$ relationship is not as strong here as the equatorial OMZs, as described above, the low oxygen would still suppress the rate of dCo scavenging in the mesopelagic where the majority of Mn oxide scavenging typically occurs (Sect. 4.4). Additionally, this region contains the transition zone chlorophyll front (TZCF), a stark transition in surface chlorophyll concentrations between the low-macronutrient subtropical gyre to the south and the iron-limited subpolar gyre to the north (Polovina et al., 2001, 2017). The TZCF seasonally migrates between 30º and 45º N, and is characterized by elevated
biological activity, a net $CO_2$ uptake flux, and a distinct $CO_2$ sink, (Juranek et al., 2012). On the GP15 expedition, prominent upper ocean and mesopelagic POC export (2000 m export depth) was also observed via $^{230}Th$ analysis in this region (Kenyon et al. in prep). High particulate export coupled with suppressed Mn oxide formation in the mesopelagic would carry biogenic particles and ligand-bound dCo associated with biomass, vitamin B$_{12}$, and/or metalloproteins into the deep ocean where the biogenic pCo is further remineralized by heterotrophs, releasing complexed dCo. The larger eukaryotes like diatoms and
coccolithophores that are supported under nutrient-replete conditions in the TZCF (Juranek et al., 2012) are also prone to faster sinking velocities (Clegg and Whitfield, 1990), which would carry this regions' biogenic particles deeper into the water column.

The remineralization of vitamin B$_{12}$ and its degradation products via eukaryotic cells could play an important role in this deep Co complexation process. Although eukaryotes cannot produce vitamin B$_{12}$, the organisms often have high vitamin
B$_{12}$ quotas (Bertrand et al., 2013). Eukaryotic cell remineralization in the mesopelagic could be supplying functionally inert Co(III) ligands in the form of these B$_{12}$ molecules to the mesopelagic and deep ocean. This process would protect the biogenic ligand-bound dCo from further scavenging, as complexed dCo is less likely to be scavenged (Saito et al., 2005), and Mn oxide precipitation rates in the deep ocean are relatively slow (Hawco et al., 2018; Johnson et al., 1996; Sunda and Huntsman, 1988). More research is needed on this deep, stable dCo fraction in the Pacific to confirm this proposed mechanism, but we believe
the unique OMZ and particle export dynamics in this region contribute to its existence.





**Figure 14: Total excess (lithogenic corrected) pCo (a), total excess (lithogenic corrected) pMn, (b), and total POC (c) transects along the GP15 cruise track. Total particulate values are summations of the small particulate, total (SPT) and large particulate, total (LPT) size fractions. Weighted-average gridding is used to interpolate between data points.**





## 4.7 Comparison to a global biogeochemical Co model


The global biogeochemical model was able to well replicate the major cycling features of dCo in the Pacific Ocean
in general, including the scavenging-induced "curl" in the dCo:dPO$_4$ relationship and the negative correlation between dCo
and O$_2$ throughout the transect (Fig. 15). It was also able to capture the major remineralization, scavenging, surface uptake,
and coastal source features observed along the GP15 transect (Fig. 16). The model was successful at replicating the major dCo
surface features across the various Pacific oceanographic provinces, appropriately deepening the nutricline and predicting
decreased dCo concentrations in the north and south subtropical gyres. The main differences between the observed and model
dCo values were due either to (1) differences in the O$_2$ and dPO$_4$ distributions that drive dCo distribution calculations, and (2)
the underestimation of some dCo flux intensities in the model.

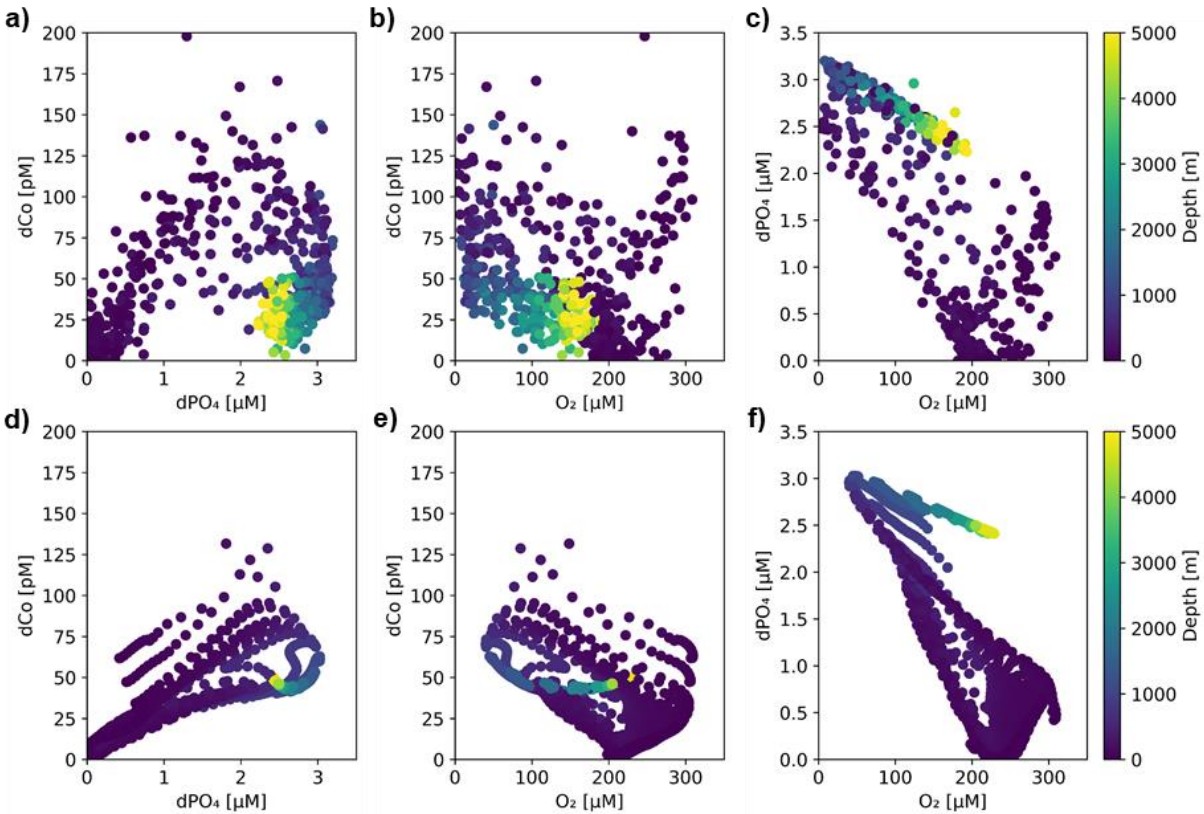

**Figure 15: Nutrient relationships between dCo, dPO$_4$ and O$_2$ in (a-c) binned observed values and (d-f) model output values. Observed
values have been binned and averaged by 1º latitude and 31 depth divisions to ensure comparability between the model and sample
values. Note that O$_2$ values are reported here in units of µM.**

The model was successful at capturing the broad, basin-wide trends across the Pacific, but did not always capture
more local processes. It did not predict the ETNP and ETSP OMZs would be present as far west as 152º W, far from the
continental coastlines, and so did not recreate the low O$_2$ or elevated dCo features observed in the GP15 transect. The model
also never reached the lowest observed O$_2$ concentrations within the North Pacific OMZ; the lowest O$_2$ concentration it

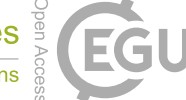

predicted in the North Pacific basin was 40 µM, while the lowest value observed within the North Pacific OMZ on the GP15
transect was 8 µM. Therefore, the biogeochemical model was able to replicate the dCo:$O_2$ correlation driven by suppressed
scavenging in OMZs across the transect, but did not capture the more intense $O_2$ minima observed and thus underestimates

dCo concentrations within the ETNP, ETSP, and North Pacific OMZs. Consistent with underestimating the extent of oxygen
depletion, the remineralization flux supplying $dPO_4$ to subsurface waters was lower in the model, shown by the positive values
of [observed $dPO_4$ – model $dPO_4$], particularly in the mesopelagic ocean. This underestimated remineralization flux also
affected the modelled dCo supply to the mesopelagic via remineralization, causing a similar underestimation of dCo values in
the model.


**Figure 16: (a-c) Model output of dCo, $O_2$, and $dPO_4$ distributions along the GP15 transect. (d-f) Differences between observed values
and model outputs of dCo, $O_2$, and $dPO_4$ concentrations along the GP15 transect. Observed values have been binned and averaged
by 1° latitude and 31 depth divisions to ensure comparability between the model and sample values. Note that $O_2$ values are reported
here in units of µM.**



Our data suggests that the dCo cycle is more robust and dynamic than the model predicts, with observations of higher dCo concentrations from sources and lower dCo concentrations from sinks than are predicted by the model. The deep ocean, for example, appears to be a larger sink of dCo via scavenging than is predicted in the model. Below 2000 m depth, the average observed dCo concentration was $27 \pm 11$ pM, but the average model concentration was consistently ~45 pM. This offset was driven by a parameterization of consistent ligand-bound dCo concentrations (Tagliabue et al., 2018), a pool of dCo that is
known to be relatively inert. In the observed transect, we do not see these levels of ligand-bound dCo in the deep ocean, with the exception of the North Pacific as described above, which suggests that the complexed dCo pool in the Pacific interior is either smaller or not as inert as the model assumed. Additionally, the source of dCo from the Alaskan coast was observed to result in greater regional dCo values, implying a larger dCo coastal source than the model predicted. This is not too surprising as coastal fluxes are often varying and unpredictable in time and location, and a global model is not expected to replicate fine-
scale regional variations of the Co cycle. However, the model included a coastal dCo source primarily from intermediate waters along the continental shelf where sediment resuspension is likely to occur, while we observed a surface dCo source primarily from coastal freshwater inputs. It is possible that the model underestimated the coastal flux of dCo from freshwater sources (see also: Bundy et al., 2020). It should be noted that the biogeochemical model was optimized for the global ocean Co cycle, and not specifically for the Pacific Ocean, which likely contextualizes much of the offset between the observed and model
transects. The model also did not incorporate a hydrothermal vent source to the Pacific, which we estimate is typically a strong local source like the one observed above the Loihi Seamount but is not a significant source to distal waters.

## 5 Conclusion

    The GP15 transect highlighted major differences in the Co biogeochemical cycle between the Pacific and Atlantic Oceans. The processes of phytoplankton uptake, remineralization, and scavenging drove much of the dCo distribution patterns
in the Pacific. Biological uptake of dCo generated a dCo:dPO$_4$ stoichiometry of $82 \pm 2$ μM:M in the upper ocean. The Pacific Co cycle was heavily influenced by the poorly-ventilated NPDW that forms a wide OMZ across the North Pacific under a high particle export region. This aged water mass displayed a weaker dCo:O$_2$ correlation than equatorial OMZs in the Atlantic and Pacific (Hawco et al., 2016; Noble et al., 2017). However, the North Pacific OMZ still appeared to suppress dCo scavenging such that elevated ligand-bound dCo was observed well below the mesopelagic. The deep, bound Co fraction was protected
from scavenging outside of the mesopelagic where most Mn-oxidizing bacteria are active. This finding represented a deviation from the expected thermohaline deep Co trend, which predicted the North Pacific would exhibit the lowest concentration of deep dCo globally due to slow but steady Co scavenging over deep ocean circulation (Hawco et al., 2018). Compared to a global ocean biogeochemical model, the observed dCo cycle (1) was more varied in distribution and (2) appeared more extreme in source and sink fluxes, indicating that the Pacific Co cycle is more dynamic than predicted by the model.

Co scavenging in the Pacific is of interest to many because of the potential for deep-sea mining in the region, especially in the Clarion-Clipperton Zone of the central North Pacific basin. Here, international mining operations to extract Fe-Mn nodules along the seafloor could being within the decade. These nodules are formed from Mn oxide scavenging over

millennia and are rich in Mn, Fe, Co, Ni, and rare earth elements (REEs) (Cameron et al., 1981; Hein et al., 2013). Co and REEs in particular are increasing in demand due to their use in personal electronic devices and sustainable energy solutions

like electric vehicle batteries and wind turbine magnets (Hein et al., 2013). However, deep-sea mining and the potential heavy-metal rich sediment plumes it could create may have serious ramifications to the relatively fragile and unexplored benthic ecosystem in the region (Drazen et al., 2020; Fuchida et al., 2017; Sharma, 2011).

This work helps to establish a baseline for dCo distributions and features in the Pacific Ocean in light of future ocean warming and anthropogenic change expected to impact this region. Relative to other trace metals, the marine dCo inventory is

among the smallest and is likely highly susceptible to regional mining and deoxygenation as anthropogenic sources of Co to the mesopelagic and deep ocean. The full scope of the impacts that expanding oxygen minima could have on metal biogeochemical cycles is still an open question, and more sophisticated modeling of the effects of deoxygenation on redox-sensitive trace elements is required.

**Code availability**

Relevant python code used for this study can be found at https://github.com/rebecca-chmiel/GP15.

**Data availability**

The GP15 expedition dCo dataset can be accessed online on the found on the Biological and Chemical Oceanography Data Management Office (BCO-DMO) website https://www.bco-dmo.org/dataset/818383 (leg 1) and https://www.bco-dmo.org/dataset/818610 (leg 2). The nutrient and hydrographic dataset can be found at https://www.bco-

dmo.org/dataset/777951 (leg 1) and https://www.bco-dmo.org/dataset/824867 (leg 2).

**Author contribution**

R. Chmiel collected and analyzed dCo samples and wrote the manuscript. M. A. Saito designed the study. M. McIlvin helped analyze dCo samples. J. Fitzsimmons, M. Hatta and N. Lanning collected and analyzed dFe and dMn samples. P. J. Lam, A. Laubach, and J.-M. Lee collected and analyzed particulate samples. W. J. Jenkins analyzed He and Ne samples. A. Tagliabue

designed and generated the geochemical model. All authors helped write the manuscript.

**Competing interests**

The authors declare that they have no conflict of interest.



**Acknowledgements**

The authors thank the crew and science party of the R/V Roger Revelle on the GP15 expedition, including Captain D. Murline
and chief scientists K. Casciotti and G. Cutter. We also thank L. Jensen, B. Summers, V. Amaral, Y Xiang, and J. Steffen for
at-sea sample collection, S. Rojas and N Carracino for POC sample processing, the Ocean Data Facility at Scripps Institution
of Oceanography for nutrient sample analysis, N. Cohen for assistance handling and transporting samples, M. Charette, S.
Dutkiewicz, and N. Hawco for science and writing insights, and D. Rao and J Saunders for python coding assistance. This
work was funded by National Science Foundation Grants OCE 1736599, 1756138, 1657781, and 1736601, and A. Tagliabue
was supported by the European Research Council (ERC) under the European Union's Horizon 2020 research and innovation
program (BYONIC, grant no. 724289).

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
