# Peer review of "Major processes of the dissolved cobalt cycle in the North and equatorial Pacific Ocean"

_Biogeosciences, 2021_

## Referee Comment (RC2)

Chmiel and coauthors present total dissolved Co and labile Co along the GP15 transect, in Pacific Ocean. As highlighted by the authors, this dataset is both one of the largest datasets to date for Co and covers a largely under-studied region, which spans distinct zones of interest for the cycling of Co and other trace metals (OMZs, hydrothermal input, etc.). I find the data to be of high quality and the interpretations well-reasoned, and I am sure this manuscript will be of interest and impactful for the community. I am happy to recommend this manuscript for publication with minor revisions, and I make the following suggestions that the authors may wish to take when revising their text.

The following is organized first into more broad discussion points/comments, followed by minor line-specific suggestions:

**Section 4.1: The impact of advection**

I think this section needs to be framed a bit more carefully to consider that, across wide depth and spatial scales where different water masses are present, the features will also be strongly influenced by advection and mixing. It is clear that this strongly impacts the basin-scale and depth profile features of other metals, rather than/in addition to 1D vertical control. This is mentioned briefly in the middle (near Line 440), but I think it would be better to start the discussion with this, and then address how other factors on top of this influence Co. To aid this, where would AAIW, NPIW and CDW fall within Figure 8? Figure 8 Panel A is useful, but in a way it assumes that all water started at the center, with uniform dCo and PO4. So that could be better contextualized by first knowing where the different water masses "start", at least at the point that they arrive in the transect

The reader could also be reminded of the short residence time of Co (at least in the surface ocean) compared to global circulation, to justify why circulation might not always be the dominant control. This topic comes up a bit in section 4.7 as well, where the "curl" in dCo v PO4 can be partly explained by water masses.

**Section 4.3: Hydrothermal sites & dCo**

The authors make use of previously reported hydrothermally-enriched metals and other tracers (Fe:$^3$He &Mn:$^3$He) to infer hydrothermal input of Co, and determine the regional importance of this. I have some hesitations regarding this interpretive structure and the final conclusions

- I am not sure the metal:$^3$He ratios presented in Jenkins et al (2020) are as well-constrained and conservative as the authors suggest (e.g., near Line 550). Jenkins et al. Fig 4 shows metal:$^3$He varying by a factor of ~2 within 100 m depth range, a similar offset observed by the authors between the apparent hydrothermal maximum in their dCo cast and the $^3$He cast. In addition, Figure 4b from Jenkins et al does not appear to show conservative Mn mixing in the plume, but rather elevated dMn independent of hydrothermal endmember (3 points near 6 nM Mn) and an open ocean endmember at low $^3$He and low Mn. In addition, high plume heterogeneity within different distinct plumes of vent sites has been reported elsewhere in the ocean (Kleint et al., 2019). As the Loihi site amalgamates different distinct venting sites into the sampled plume, shifting position in the plume might be expected to alter the exact geochemical conditions. One more point, though I haven't seen this published, so maybe it is of limited use. But I mention it to note that others in the field are trying to solve this problem, and I'm not sure there is a clear solution yet - Maeve Lohan presented a talk at Ocean sciences 2020 discussing the difficulties in matching standard CTD $^3$He and trace metal casts (CT31A-01, but the abstract doesn't cover much of this).

  I suggest revising the assumption of consistent metal:$^3$He across different specific venting points and making the discussion a bit more conservative. The uncertainty in reconstructing Co fluxes should be revised to include the extrapolated uncertainty/variability in Me:$^3$He. I think this section of the text still contains important information, as does the manuscript as a whole, so making this less quantitative should not alter the value of the section or manuscript overall.

- The dCo:$^3$He regression (Figure 11) reaches zero $^3$He above the general deep Pacific background dCo (regression dCo ~2x deep Pacific dCo). Why is that? Does this indicate a second potential source of Co (e.g. Co scavenged from the plume, delivered to nearby sediments, and then an elevated flux of dCo out of vent-adjacent sediments?

- It seems the GP15 transect as a whole did not track the Loihi hydrothermal plume well. Based on Jenkins et al. (2020), the only stations that seem within the plume are 18, 19, PR & LS (their Figure 5). Elevated dCo is observed at stations LS, PR & 19, and station 18 also shows elevated lCo (and maybe also a bit of elevated dCo). The absence of hydrothermal Co outside of these stations seems expected based on the results of Jenkins et al indicating the absence of hydrothermal influence elsewhere along the transect. Also, the plume maximum depth (1100) wasn't sampled at other stations. Therefore, I don't think the necessary data are available to speak to the longevity of dCo within the plume, either for or against it. Rather, it seems that the cruise simply didn't sample the more distant plume. Again, I don't think revising the text in this way alters the significance of the section or text.

**Sections 4.4 & 4.5: dCo in OMZs and implications under OMZ expansion**

The authors assess the role of OMZs in enriching dCo, including variability within different OMZ regions, and quantitatively estimate the impact for future Co cycling. I found this section quite interesting, and I have the following suggestions for the interpretive framework and quantifications.

- OMZ dCo-$O_2$ regressions (Figure 12 and elsewhere): I don't think this [$O_2$] window does a good job of isolating the affects of OMZs. The plots are dominated by the behavior of dCo in largely oxic waters, which are mostly widely removed from OMZ waters. This is also visible in the dCo v $O_2$ distribution, which is non-linear across this range and instead seems to show a steep trend at low $O_2$ and a shallow trend at higher $O_2$, while dCo largely plots below the linear regressions in the middle of the range. The Hawco et al., 2016 data also show this. Therefore, the derived slope may be strongly biased by mixing/circulation, and may not give the most accurate information for projecting Co accumulation with decreasing $O_2$.

  Mn redox cycling also supports the choice of a different $O_2$ window – if the suppression of Mn oxide formation and reduction of Mn oxides is an important factor for dCo, then the range of $O_2$ values should reflect what is most relevant for dMn. This probably isn't from 200-100 µM, and instead is maybe <100 µM $O_2$ (or maybe even lower – the kink/bend in the plots seems to fall near 50 µM $O_2$, with possibly a second one near 100). Therefore, I think a more selective approach focusing on OMZ and immediately adjacent waters would give more meaningful results here.

- Differences across different OMZs: I appreciated the discussion of differences between the OMZs observed along this transect. I think this is important in understanding and projecting Co cycling as a whole. I think this discussion could benefit from better incorporation of our understanding of the cycling of other trace metals in these environments, and specifically those most relevant to Co, to strengthen a mechanistic understanding.
  - The trend between Co and $O_2$ is less meaningful in the subarctic OMZ. Data for dissolved Mn here suggest that the dMn peak is not at the $O_2$ minimum (e.g. Martin et al., 1989; Sim & Orians, 2019). Therefore, I am not sure that a priori assumption should be a coherent linear relationship between dCo and $O_2$. Instead, it would involve one factor contributing to a dCo peak above the $O_2$ minimum from pMn reduction, and a second in the $O_2$ minimum from the respiration of biogenic material. And in this case, across a wide OMZ, perhaps the assumption of a linear trend isn't as applicable. This is addressed somewhat, but I think could be strengthened by bringing in more of this sort of supporting data. Ohnemus et al. (2019, see, for example, Table 3) might be relevant here for building that interpretive framework, and also could be helpful elsewhere in the manuscript.

- There is elevated pMn surrounding OMZs. How might this affect the potential of dCo to escape OMZ settings?

**Other general comments**

Detection limits & data treatment

- I realize that it is tough to define a detection limit off of n = 2 blanks, but choosing the only blank with n = 3 to define the detection limit results in one of the lower possible detection limits. That optically isn't the best I think, and I think it would be better to use variable detection limits throughout the cruise, with samples compared against the relevant set of reagents for that analysis.
- Lines 172-173: I appreciate the authors being careful here, but the uncertainty ranges of these two overlap and therefore I'm not sure it makes sense to discuss them as different.
- Line 343, 380: It should be noted here that these values are below the detection limit, and maybe it is better not to quote a number for concentrations below the detection limit since that number isn't meaningful. This shows up a couple other places in the manuscript as well (e.g. labile dCo in deep water).

Colloidal Co: Throughout the text it is assumed that the difference between labile and dCo is due to strong organic complexes. Perhaps I'm a bit ignorant on this, but do we know to what extent colloidal Co would show up as labile v. total dissolved? If that's known, it might be good to remind the readers of this somewhere. And, if it's not known, maybe colloidal Co could be important in some sites (e.g. the hydrothermal vent site, where other metals are known to be enriched in colloidal phases). This is also where labile:dCo is different than throughout the transect.

Section 4.6 depth windows: I don't think the depth integration here, giving units of Co per m$^2$, is the best approach. This could be strongly biased by variable depth: at the equator, the depth range integrated here is about 3000 m, while is is about 6000 m at 12 N. I think turning this depth integrated value back into an average (i.e. a depth-weighted average) would give a more meaningful comparison. Likewise, starting this window at 1000 m may be problematic, as it catches the core of the N Pacific OMZ but is below the OMZ in the subtropical Pacific. These OMZ-specific trends already have two discussion sections, and here the focus seems to be more on the stable dCo in deeper waters. Maybe starting the depth window at 2000 m would better isolate trends in deep water.

**Line-specific comments**:

Line 19: Approximately what depth range counts as the "upper ocean"

Line 20: I suggest using mol:mol ratios rather than mol/l:mol/l

Line 24: Later in the text "potential xs$^3$He" is used rather than "estimated xs$^3$He". It might be better to consistently use only one.

Lines 59-60: I suggest "species" instead of "speciation".

Lines 143-147: I think it would be useful here to say slightly more about this offset – specify which method gives higher and which gives lower, and why that might be. Or if it is not consistent, that could also be stated.

Lines 236-237 & Lines 536-548: I think small movements in ship position and/or heterogeneity in the hydrothermal plume should be considered. A thousandth of a decimal degree is about 100 m, the same distance quoted as possible due to wire angle. Also, I think Lines 536-548 could be shortened a bit, to

simply say that the casts seemed to catch different parts of the plume, and given the depth and local heterogeneity, many different explanations may exist for why they were different.

Lines 365-374: It might be more useful to compare with the Southern Ocean instead of the Atlantic, since the Southern Ocean is more proximally the source of North Pacific deep water.

Line 477: It is a bit ambiguous what the respectively refers to, and from my reading I would think it was slope error and $R^2$ rather than stations 2 and 3 (but what is shown appears to be R2 for stations 2 and 3).

Lines 492-495: The Peru margin is quite a different setting (strong OMZ), and therefore might not be the most directly applicable to this site. Data from an oxic setting might be more useful for a comparison.

Lines 499-503: de Carvalho et al (2021) could be a useful reference here.

Line 525: Does "Co distributions" refer to dCo or labile dCo or pCo, or all three?

Line 528: "Metallic" sounds a bit funny to me.

Line 588: I'm not sure I understand why a negative correlation is expected. If there is a dCo source from the vent, wouldn't a positive correlation be expected?

Line 682: It's not clear to me what a volumetric width is. Maybe the calculation can also be described in a bit more detail (perhaps in a supplement, with an example figure)

Lines 726-746: These two paragraphs are a bit redundant, and can maybe be combined into one paragraph.

Lines 806-808: It is probably both of these, and not just one or another, as the rest of the discussion section and figures identify specific examples of these two. In that respect, I think it is useful to have this short discussion section addressing the models, but I think it could be streamlined a bit. It is hard to have a detailed discussion of the differences between models and this dataset without explaining the model in more detail (which isn't the purpose of this manuscript). I recommend spending less time discussing detailed differences between the model and the new data, and center the discussion on highlighting how this dataset can aid future models improve based on the new constraints on the Co cycle discovered here. This mostly applies to the final paragraph, for example where the initial presentation suggests that dCo is more scavenged in the Pacific than expected (in contradiction to the earlier discussion), but rather it is just that the model used a different parameterization than these new data would suggest (i.e. opportunity to improve the model).

Lines 860-868: I think it is reasonable to include this, but the transitions to and from this section are a bit abrupt, and this topic wasn't addressed anywhere else in the text. Maybe it could be moved or somehow incorporated better, so that it doesn't break the flow of points before and after (as built up by the rest of the manuscript).

General comment on figures: It would be nice if the figures could be a bit more internally consistent, and a bit more easy to differentiate. A generally consistent color scheme is used throughout the figures, but this is applied inconsistently to the parameters plotted. For example, Figure 5 uses teal to mean dissolved Co while the other figures use blue to signify this, and teal to indicate labile cobalt.

Figures 3 & 5: I think adding station numbers above the top panel would be helpful.

Figure 4: It would be good to include visual color legend somewhere, even if it is a bit intuitive. Also, I find this figure a bit hard to read. Maybe the figure would be easier to read if the surface box was somehow expanded (either it took more space, or it covered 0-500 m instead of 0-1000). Also, it would be nice if the scaling for the upper 1000 m was the same for the deep stations as it is for the partial stations (7, 9, 13, 15, 17, 20, 22, 34, 36, 38).

Figure 5: It would be nice to include the latitudinal zones here, as with figure 3. It also seems like the Loihi Seamount is missing here. Also, the order of panels ($PO_4$ and $O_2$ specifically) is reversed compared to Figure 3. Finally, maybe this could be combined with Figure 3 to make an 8 panel figure.

Figure 6: The y scaling is different in each of these three panels. Maybe the bottom two could use the same range.

Figure 7: I don't think the 1:1 line actually shows a 1:1 relationship. It looks like 120 pM labile Co is less than 100 pM total dCo on this line.

Figure 9: Panel J is a great addition! Maybe the other panels could be improved by writing the approximate depth covered by this isopycnal range in parenthesis after the isopycnals. This can be re-created by going back and forth between the panels, but having it written would be nice.

Figure 10: Please use different colors for Mn, Ti and Al compared to each other and what is used for Co. Also, the ranges of LPT and SPT are different, but this isn't obvious and it gives the impression that LPT is approximately the same as SPT. If it is necessary to use different ranges, please make this more clear, and maybe differentiate the ranges so the data don't overlap.

Figure 11: Here dCo is presented as purple (panel A), compared to blue in most of the other plots.

Figure 13: I don't see corrected pCo on panel C. The color choice here could create some confusion with labile dCo. Also, it might be nice if Panel D didn't use the Co color scheme, and if the different ranges could be somehow better differentiated (as with Figure 10). As it is now, it gives the initial impression that LPT POC = SPT POC most of the time.

**References**:
de Carvalho, L.M., Hollister, A.P., Trindade, C., Gledhill, M. and Koschinsky, A., 2021. Distribution and size fractionation of nickel and cobalt species along the Amazon estuary and mixing plume. *Marine Chemistry*, *236*, p.104019.
Jenkins, W.J., Hatta, M., Fitzsimmons, J.N., Schlitzer, R., Lanning, N.T., Shiller, A., Buckley, N.R., German, C.R., Lott III, D.E., Weiss, G. and Whitmore, L., 2020. An intermediate-depth source of hydrothermal 3He and dissolved iron in the North Pacific. *Earth and Planetary Science Letters*, *539*, p.116223.
Martin, J.H., Gordon, R.M., Fitzwater, S. and Broenkow, W.W., 1989. VERTEX: phytoplankton/iron studies in the Gulf of Alaska. *Deep Sea Research Part A. Oceanographic Research Papers*, *36*(5), pp.649-680.
Ohnemus, D.C., Torrie, R. and Twining, B.S., 2019. Exposing the distributions and elemental associations of scavenged particulate phases in the ocean using basin-scale multi-element data sets. *Global Biogeochemical Cycles*, *33*(6), pp.725-748.
Sim, N. and Orians, K.J., 2019. Annual variability of dissolved manganese in Northeast Pacific along Line-P: 2010–2013. *Marine Chemistry*, *216*, p.103702.

---

## Author Comment (AC1)

Author Response to Anonymous Referee #1

Referee comment on "Major processes of the dissolved cobalt cycle in the North and equatorial Pacific Ocean" by Rebecca Chmiel et al., Biogeosciences Discuss., https://doi.org/10.5194/bg-2021-305-RC1, 2021

This is a review of the paper titled "Major processes of the dissolved cobalt cycle in the North and equatorial Pacific Ocean" by Rebecca Chmiel et al. for consideration in Biogeosciences Discussions. The authors show high quality dissolved and labile cobalt data from a recent GEOTRACES expedition (GP15) to the North and South Pacific Oceans. Co shows evidence of biological uptake and remineralization in the surface ocean and scavenging with depth, accordant with its typical global distribution. While hydrothermal sources were observed and discussed, persistent oxygen minimum zones sampled in this study were determined to be a larger input of dissolved Co to intermediate waters. The distribution of labile vs organically-complexed Co, as determined by UV radiation, is discussed throughout, highlighting in particular the importance of ligands in protecting dCo in Pacific deep waters. Comparison to particulate samples, collected by pumping, and Fe and Mn are also discussed at select stations.

I found this manuscript to be an enjoyable and easy read. While these processes are being explored across multiple elements for other GEOTRACES parameters, this manuscript provides a nice foundation for oxidative release, scavenging, and mixing of deep waters along this transect. The organization supported the findings and the logic of conclusions throughout. I thought the addition of other dissolved metals (Fe and Mn) and particle species (pCo, POC, etc) to be complimentary rather than extraneous in telling the story. The figures were likewise appropriate and well-designed.

I recommend acceptance to Biogeosciences following minor revisions as outlined below. General comments to this manuscript: While there was thorough discussion of possible processes it felt a little long and I was having trouble following the deep water/POC export and modeling sections as a result. I wonder if there is a way to streamline? It seems that you discuss deep water Co trends through the lens of Co:PO4, labile Co, and then again in a separate section. I also thought that the mention in the conclusions about deepwater Fe-Mn nodules as a source of Co could be expanded slightly. Do you expect this inventory to increase with increased deoxygenation, for example? How efficient are Fe-Mn nodules in scavenging Co? Just thinking this would be of interest to a broader community and inform ongoing deepwater mining studies.

- Thank you for your review and comments. It is good to hear that despite the manuscript's length, it was an enjoyable read. We have organized the paper by relevant processes and not by oceanographic location, and so restructuring the deep water discussion to one section of the manuscript would unfortunately not aid reader comprehension.

  Although the question of how deep dCo will change with deoxygenation is an interesting one, it is not one we are able to answer with the simple estimation performed in section 4.5. Any further analysis of the effects of deoxygenation on Pacific Co would likely require a dedicated modeling study, which we believe to be a very compelling idea for future work.

  We have also slightly expanded the paragraph on seafloor mining in the conclusion to include a clearer description of the Fe-Mn crusts and Mn nodules, and their importance to the geoscience community. Neither actively scavenge dCo – instead, dCo is scavenged to pCo via Mn-oxide

formation in the water column, and the sinking oxides slowly build the nodule along the seafloor via accretion. This paragraph now reads:

"This study improves our understanding of dCo dynamics in the Pacific water column, but it should be mentioned that many are interested in Pacific Co biogeochemical cycling as it relates to Co scavenging and mineralization along the seafloor and the emerging deep sea mining industry. When dCo is scavenged and removed from the water column in the form of pCo incorporated into Mn-Oxide particles, it sinks and forms vast regions of ferro-manganese oxide crusts embedded with metal-rich nodules that accrete slowly along the seafloor (Aplin and Cronan, 1985). These nodules are formed from Mn oxide deposition over millennia and are rich in Co, Ni, Cd, Zn and rare earth elements (REEs) (Cameron et al., 1981; Hein et al., 2013). Such deposits are of interest to many because of the potential for deep-sea mining in the region, especially in the Clarion-Clipperton Zone of the central North Pacific basin. Here, international mining operations to extract Fe-Mn nodules along the seafloor could being within the decade. The value of Co ore, which is currently only mined terrestrially, has been increasing in demand over the past two decades due to the metal's use in personal electronic devices and sustainable energy solutions like electric vehicle batteries, and many believe that mining the vast, concentrated deposits of Co on the Pacific seafloor could alleviate this demand (Hein et al., 2013). However, deep-sea mining and the potential heavy-metal rich sediment plumes it could create may have serious ramifications to the relatively fragile and unexplored benthic ecosystem in the region (Drazen et al., 2020; Fuchida et al., 2017; Sharma, 2011). Although the technology and facilities required to mine the deep sea have proven to be both complicated and expensive (Cameron et al., 1981), the potential new source of Co ore could, in theory, help support widespread adoption of electric vehicle fleets (Hein et al., 2013)."

Line comments:

Line 121: "The Chelex resin was prepared as described…"

- This sentence now reads "The Chelex resin was prepared as described…"

Line 135: Can you clarify if the labile dCo analyses were done on the exact same sample as the total dCo? Or an aliquot? If I understand correctly, the total dCo was done using an aliquot, does that mean that the rest of the sample is only able to be used for repeat labile analyses in the future?

- Both labile and total dCo samples were aliquoted from the same 60 mL sample bottle, and this has been clarified in the text, which now reads "… 11 mL of sample seawater was aliquoted from the sample bottle into 15 mL acid-washed polypropylene vials…"

Lines 495-496: I wonder if this was true in the Arctic as well, since you just compared the Alaskan source to Arctic processes?

- It was! The Arctic Ocean also shows a negative correlation between dCo and salinity within the Transpolar Drift. This comparison has been added.

Line 501: What are typical river endmember concentrations of Co in this area?

- There's little published data of dCo from rivers in this region. A GEOTRACES–affiliated fieldwork team from the Charette and Fitzsimmons labs collected riverine and estuarine samples from the Alaskan coast near station 1 that are currently unpublished, showing ICP "labile" dCo data around 4-18 nmol/kg. However, as was discussed in the labile methods section, this is not directly comparable to electrochemical dCo data. For this reason, we have not included observed endmember values for Alaskan riverine dCo, but we are confident our endmember estimate of ~13 nM dCo is a very reasonable estimate.

Line 505: By "open" do you mean open ocean or yet unknown?

- Open ocean - this has been clarified and now reads "… is mixing with an open ocean endmember that contains…"

Line 536: ODF stands for "Oceanographic Data Facility", I believe this is wrong in your acknowledgements as well. I would mention it is hosted at Scripps Institution of Oceanography.

- This correction has been made. The connection to Scripps was already mentioned in the macronutrients methods section (Sect. 2.1) and in the acknowledgements.

Line 541: Can you clarify which samples are coming from the ODF rosette? The FIA Fe and Mn samples?

- This was explained in the methods section 2.6, but additional clarification has been added in the discussion section to make the distinction clear to readers. The section now reads "…the dFe:xs$^3$He and dMn:xs$^3$He ratios from the ODF cast, measured via FIA,…" and "…the GTC profile dFe and dMn values, measured via ICP-MS,…"

Line 578+595-596: It is odd to mention the flux from hydrothermal vents twice with different units. Can you combine these sentences?

- The second mention of the global Co hydrothermal flux has been removed.

Line 591: local scavenging of the labile or ligand-bound Co?

- Labile Co – this has been clarified and now reads "…likely due to local scavenging of labile dCo before it is transported…"

Line 749: "expeditions"

- This spelling correction has been made.

Lines 843-845: Will this model now be updated to reflect these new Co data?

- The discussion of model data as it compares to the observed data has been updated to reflect more on the future of the model and how this comparison will be used to improve upon future iterations of the model. The final paragraph of this section now reads:

  "Recognizing inconsistencies between expected model dCo distributions and observed transect dCo distributions is an effective way to test and improve upon the biogeochemical model's parameters. The parameterization of ligand-bound cobalt in the deep Pacific, for example, could

be improved by decreasing the assumed concentration of inert, organically bound dCo in deep water masses. It should be noted that the biogeochemical model was optimized for the global ocean Co cycle, and not specifically for the Pacific Ocean, which likely contextualizes much of the offset between the observed and model transects, particularly for inconsistencies within the $O_2$ and $PO_4$ distributions. The model also did not incorporate a hydrothermal vent source to the Pacific, which we concur is an appropriate omission on a basin scale. Future iterations of the Co biogeochemical model will consider the successes and inconsistencies presented here and continue to improve upon our conceptualization and parameterization of the marine cobalt cycle."

---

## Author Comment (AC2)

Author Response to David Janssen (Referee)

Referee comment on "Major processes of the dissolved cobalt cycle in the North and equatorial Pacific Ocean" by Rebecca Chmiel et al., Biogeosciences Discuss., https://doi.org/10.5194/bg-2021-305-RC2, 2022

Chmiel and coauthors present total dissolved Co and labile Co along the GP15 transect, in Pacific Ocean. As highlighted by the authors, this dataset is both one of the largest datasets to date for Co and covers a largely under-studied region, which spans distinct zones of interest for the cycling of Co and other trace metals (OMZs, hydrothermal input, etc.). I find the data to be of high quality and the interpretations well-reasoned, and I am sure this manuscript will be of interest and impactful for the community. I am happy to recommend this manuscript for publication with minor revisions, and I make the following suggestions that the authors may wish to take when revising their text.

The following is organized first into more broad discussion points/comments, followed by minor line-specific suggestions:

- Thank you for your review and comments. They were thorough, knowledgeable, and very helpful.

**Section 4.1: The impact of advection**

I think this section needs to be framed a bit more carefully to consider that, across wide depth and spatial scales where different water masses are present, the features will also be strongly influenced by advection and mixing. It is clear that this strongly impacts the basin-scale and depth profile features of other metals, rather than/in addition to 1D vertical control. This is mentioned briefly in the middle (near Line 440), but I think it would be better to start the discussion with this, and then address how other factors on top of this influence Co. To aid this, where would AAIW, NPIW and CDW fall within Figure 8? Figure 8 Panel A is useful, but in a way it assumes that all water started at the center, with uniform dCo and PO4. So that could be better contextualized by first knowing where the different water masses "start", at least at the point that they arrive in the transect

The reader could also be reminded of the short residence time of Co (at least in the surface ocean) compared to global circulation, to justify why circulation might not always be the dominant control. This topic comes up a bit in section 4.7 as well, where the "curl" in dCo v PO4 can be partly explained by water masses.

- To improve and centralize the discussion of dCo advection, a brief paragraph describing the effects of advection on dCo distributions has been added to the beginning of the discussion section, and section 4.1 has been renamed "The dCo and dPO$_4$ relationship depicts Co advection, uptake, remineralization, and scavenging." This new paragraph states:

  "The advection of dCo is an important driver of dCo distributions throughout the Pacific. In the deep ocean, CDW carries dCo north from the Southern Ocean to the North Pacific basin via thermohaline circulation, then partially upwells the dCo in the North Pacific basin, forming NPDW. To either side of the equator, dCo is advected at intermediate depths within the ETNP and ETSP OMZs by the North and South Equatorial currents, carrying elevated concentrations of dCo from the continental margins to the pelagic Pacific Ocean (see Sect. 4.4) (Hawco et al., 2016). However, dCo's relatively short residence time, especially in the upper ocean and

mesopelagic, allows other, more vertical processes such as biological uptake, remineralization and scavenging to overlap with the dCo advection signal. This overlap of horizontal (advection) and vertical processes determines the Pacific dCo distribution observed along the GP15 transect."

Approximate positions of CDW and PDW are now labeled on Figure 8b to clarify the advection-driven curl towards the origin. The dCo values of the water mass endmembers (the "start" of the vector space) is not currently known and beyond the scope of this study, but we hope labelling the water mass positions will aid reader comprehension.

**Section 4.3: Hydrothermal sites & dCo**

The authors make use of previously reported hydrothermally-enriched metals and other tracers (Fe:3He &Mn:3He) to infer hydrothermal input of Co, and determine the regional importance of this. I have some hesitations regarding this interpretive structure and the final conclusions

- I am not sure the metal:3He ratios presented in Jenkins et al (2020) are as well-constrained and conservative as the authors suggest (e.g., near Line 550). Jenkins et al. Fig 4 shows metal:3He varying by a factor of ~2 within 100 m depth range, a similar offset observed by the authors between the apparent hydrothermal maximum in their dCo cast and the 3He cast. In addition, Figure 4b from Jenkins et al does not appear to show conservative Mn mixing in the plume, but rather elevated dMn independent of hydrothermal endmember (3 points near 6 nM Mn) and an open ocean endmember at low 3He and low Mn. In addition, high plume heterogeneity within different distinct plumes of vent sites has been reported elsewhere in the ocean (Kleint et al., 2019). As the Loihi site amalgamates different distinct venting sites into the sampled plume, shifting position in the plume might be expected to alter the exact geochemical conditions. One more point, though I haven't seen this published, so maybe it is of limited use. But I mention it to note that others in the field are trying to solve this problem, and I'm not sure there is a clear solution yet - Maeve Lohan presented a talk at Ocean sciences 2020 discussing the difficulties in matching standard CTD 3He and trace metal casts (CT31A-01, but the abstract doesn't cover much of this).

  I suggest revising the assumption of consistent metal:3He across different specific venting points and making the discussion a bit more conservative. The uncertainty in reconstructing Co fluxes should be revised to include the extrapolated uncertainty/variability in Me:3He. I think this section of the text still contains important information, as does the manuscript as a whole, so making this less quantitative should not alter the value of the section or manuscript overall.

  - This is an interesting point, and the Kleint article provided a great perspective. The discussion of the assumptions of consistent trace metal:xs$^3$He ratios has been re-written to make the potential xs$^3$He results more conservative. Additionally, we have added error bars showing the propagation of error from the regressions in Fig. 11f to all plots showing the potential xs$^3$He value (Fig. 11e, g). However, we believe that a quantitative analysis of the dCo flux, despite the calculation's caveats, is both relevant and significant. To our knowledge, this is the first estimation of dCo flux from hydrothermal vent fluid on a local water-column scale (i.e. not directly from the vent fluid elevated through the water column.)

- The dCo:3He regression (Figure 11) reaches zero 3He above the general deep Pacific background dCo (regression dCo ~2x deep Pacific dCo). Why is that? Does this indicate a second potential source of Co (e.g. Co scavenged from the plume, delivered to nearby sediments, and then an elevated flux of dCo out of vent-adjacent sediments?

  - The dCo hydrothermal signal has a positive y-intercept because, unlike dFe and dMn, hydrothermal dCo flux supplies dCo to the above water column on the same order of magnitude as other oceanographic processes that dictate the cobalt cycle, in this case, remineralization in the mesopelagic. This is in contrast with dFe and dMn, whose hydrothermal signal dwarfs their mesopelagic and surface ocean distributions, and thus hydrothermalism controls/drives the distribution of dFe and dMn at this station more than the distribution of dCo.

- It seems the GP15 transect as a whole did not track the Loihi hydrothermal plume well. Based on Jenkins et al. (2020), the only stations that seem within the plume are 18, 19, PR & LS (their Figure 5). Elevated dCo is observed at stations LS, PR & 19, and station 18 also shows elevated lCo (and maybe also a bit of elevated dCo). The absence of hydrothermal Co outside of these stations seems expected based on the results of Jenkins et al indicating the absence of hydrothermal influence elsewhere along the transect. Also, the plume maximum depth (1100) wasn't sampled at other stations. Therefore, I don't think the necessary data are available to speak to the longevity of dCo within the plume, either for or against it. Rather, it seems that the cruise simply didn't sample the more distant plume. Again, I don't think revising the text in this way alters the significance of the section or text.

- Figure 6 of Jenkins et al. (2020) shows xs³He, dFe, and dMn concentrations at distal stations interpolated to 1100 depths. This linear interpolation was necessary because the plume depth of 1100 was not sampled at these distal stations, as you noted. Jenkins' analysis shows a clear negative relationship between the three plume-associated species and distance from the Loihi station. In contrast, when we repeated this analysis with dCo and labile dCo, essentially no relationship existed.

  This paragraph has been revised to be more clear and conservative in its conclusions. We note that the cruise track did not capture the full extent of the plume, and we have streamlined the analysis. The main focus is now that dCo acts differently than dFe and dMn at distal stations, and that it is likely not transported very far. The paragraph now reads:

  "The Loihi Seamount was the source of an eastward-flowing distal plume of elevated xs³He that stretched across an estimated $1×10^{12}$ m² area of the Pacific basin (Jenkins et al., 2020). Even though the GP15 transect did not capture the full extent of this hydrothermal plume, xs³He, dMn and dFe were found to be elevated at nearby stations (18, 19, and the Puna Ridge); when interpolated to the core plume depth of 1100 m, dFe and dMn showed a positive correlation with xs³He at distal stations (Jenkins et al., 2020, Figure 6). In contrast, dCo concentrations interpolated to 1100 m from distal stations (14-25) showed little correlation with xs³He ($R^2$ = 0.28), indicating that hydrothermally-associated dCo was not likely to have been transported far within the plume. Although it is difficult to make assumptions about a plume that was not thoroughly sampled by the expedition track, this finding is consistent with the hypothesis that the majority of hydrothermally-sourced dCo is scavenged before it is able to be transported far on a basin-scale."

**Sections 4.4 & 4.5: dCo in OMZs and implications under OMZ expansion**

The authors assess the role of OMZs in enriching dCo, including variability within different OMZ regions, and quantitatively estimate the impact for future Co cycling. I found this section quite interesting, and I have the following suggestions for the interpretive framework and quantifications.

- OMZ dCo-O2 regressions (Figure 12 and elsewhere): I don't think this [O2] window does a good job of isolating the affects of OMZs. The plots are dominated by the behavior of dCo in largely oxic waters, which are mostly widely removed from OMZ waters. This is also visible in the dCo v O2 distribution, which is non-linear across this range and instead seems to show a steep trend at low O2 and a shallow trend at higher O2, while dCo largely plots below the linear regressions in the middle of the range. The Hawco et al., 2016 data also show this. Therefore, the derived slope may be strongly biased by mixing/circulation, and may not give the most accurate information for projecting Co accumulation with decreasing O2.

  Mn redox cycling also supports the choice of a different O2 window – if the suppression of Mn oxide formation and reduction of Mn oxides is an important factor for dCo, then the range of O2 values should reflect what is most relevant for dMn. This probably isn't from 200-100 μM, and instead is maybe <100 μM O2 (or maybe even lower – the kink/bend in the plots seems to fall near 50 μM O2, with possibly a second one near 100). Therefore, I think a more selective approach focusing on OMZ and immediately adjacent waters would give more meaningful results here.

- Differences across different OMZs: I appreciated the discussion of differences between the OMZs observed along this transect. I think this is important in understanding and projecting Co cycling as a whole. I think this discussion could benefit from better incorporation of our understanding of the cycling of other trace metals in these environments, and specifically those most relevant to Co, to strengthen a mechanistic understanding.

  - The trend between Co and O2 is less meaningful in the subarctic OMZ. Data for dissolved Mn here suggest that the dMn peak is not at the O2 minimum (e.g. Martin et al., 1989; Sim & Orians, 2019). Therefore, I am not sure that a priori assumption should be a coherent linear relationship between dCo and O2. Instead, it would involve one factor contributing to a dCo peak above the O2 minimum from pMn reduction, and a second in the O2 minimum from the respiration of biogenic material. And in this case, across a wide OMZ, perhaps the assumption of a linear trend isn't as applicable. This is addressed somewhat, but I think could be strengthened by bringing in more of this sort of supporting data. Ohnemus et al. (2019, see, for example, Table 3) might be relevant here for building that interpretive framework, and also could be helpful elsewhere in the manuscript.

- It is stated in the manuscript that the negative correlation between O₂ and dCo should be interpreted as mixing line, especially in the ETNP and ETSP where advection from the Eastern boundary currents along the Americas supplies coastal dCo to the pelagic Pacific. We have re-written this section to clarify that dCo scavenging suppression likely only occurs at very low O₂ concentrations in the cores of the OMZs, and to highlight the role of mixing and circulation.

We have removed the whole-transect dCo:$O_2$ relationship (previously Fig. 12a) as it showed a non-relevant correlation ($R^2 = 0.22$), had little to contribute to the manuscript's narrative, and distracted from the role of dCo within specific OMZs, which is a much more compelling discussion. We have also made the discussion of the North Pacific OMZ more qualitative and removed the correlation line from its dCo:$O_2$ subplot; we believe a more descriptive analysis of this relationship is more useful than fitting it to a poorly-correlated linear regression. There are many processes overlapping in the North Pacific, and we've updated this analysis to better describe them (see below).

We have also lowered the $O_2$ window for the dCo vs. $O_2$ correlations from 200 µmol kg$^{-1}$ to 150 µmol kg$^{-1}$, which raised the $R^2$ values of the ETNP and ETSP two-way regression lines from 0.51 and 0.59 to 0.66 and 0.72, respectively. This change in the $O_2$ window removed a flattening of the dCo:$O_2$ slope that occurred between 150-200 µmol kg$^{-1}$ $O_2$. The new dCo:$O_2$ slopes are more negative than we originally presented, and so changed the deoxygenation calculations presented in Section 4.5; The new estimations for a changing dCo inventory suggest a larger rate of change ($\Delta dCo/\Delta t$) and a larger % increase of the dCo inventory. The new analysis is presented in Table 4 (previously Table 3):

A discussion of the dCo:$O_2$ relationship at a lower $O_2$ window is a compelling one, and we explored this analysis in greater depth according to your recommendation. We will include this analysis as an appendix to the manuscript, which is also included at the end of this document (Appendix A). Briefly, in the ETNP and ETSP, the dCo:$O_2$ slope does become more vertical (more negative) at an $O_2$ window of < 50 µmol kg$^{-1}$, but the correlation is not very significant ($R^2 = 0.03$, 0.09 and 0.47 in the North Pacific, ETNP and ETSP, respectively). The higher slopes occur at shallower depths close to the 150 m depth threshold, and we believe they are due to remineralization and scavenging processes, especially scavenging at the boundaries of the OMZ core. This more negative slope thus represents more vertical processes overlapping with the horizontal process of advection and mixing. The nonlinear relationship is most pronounced in the North Pacific, and so we have separated these two sections of the North Pacific dCo:$O_2$ relationship in the new Fig. 12a and included a qualitative analysis of this trend in the paper text.

- There is elevated pMn surrounding OMZs. How might this affect the potential of dCo to escape OMZ settings?

  - pMn concentrations have proved difficult to use to directly trace scavenging processes because they tend to be removed from the water column relatively quickly. A more in-depth analysis of pMn is outside the scope of this study.

**Other general comments**

Detection limits & data treatment

- I realize that it is tough to define a detection limit off of n = 2 blanks, but choosing the only blank with n = 3 to define the detection limit results in one of the lower possible detection limits. That optically isn't the best I think, and I think it would be better to use variable detection limits throughout the cruise, with samples compared against the relevant set of reagents for that analysis.

- o This dataset, including the calculation of the detection limit, has been approved by the GEOTRACES intercalibration committee. Of the 3 other standard deviations that could be used to calculate the detection limit, 2 are very similar to the standard deviation used (one slightly higher and one slightly lower) and would not change the detection limit by a substantial amount. We left this section as is.

- Lines 172-173: I appreciate the authors being careful here, but the uncertainty ranges of these two overlap and therefore I'm not sure it makes sense to discuss them as different.

  - o The discussion of the GSC2 standard has been edited to highlight the overlapping uncertainties of the values, and no longer implies they are statistically different. It now reads "The D1 standard (44.0 ± 0.01 pM, n = 2) was found to be within one standard deviation of the consensus value (46.6 ± 4.8 pM), and the GSC2 standard (82.8 ± 2.9, n = 4) overlaps in its uncertainty range with the value reported in (Hawco et al., 2016) (77.7 ± 2.4, 6.2 % difference)."

- Line 343, 380: It should be noted here that these values are below the detection limit, and maybe it is better not to quote a number for concentrations below the detection limit since that number isn't meaningful. This shows up a couple other places in the manuscript as well (e.g. labile dCo in deep water).

  - o We disagree that the samples with cobalt concentrations below the detection limit are not meaningful. The true values of dCo are very small – if there's significant concentrations of organic ligands, the labile concentrations will be incredibly low – and measuring a value below the detection limit is meaningful. Our dataset of high-throughput values for trace metal speciation is rare, and removing these values from our analysis misrepresents that data and would skew our results to appear higher than they really are. We are clear and transparent about our treatment of results measured below the detection limit, and we left these sections as is.

Colloidal Co: Throughout the text it is assumed that the difference between labile and dCo is due to strong organic complexes. Perhaps I'm a bit ignorant on this, but do we know to what extent colloidal Co would show up as labile v. total dissolved? If that's known, it might be good to remind the readers of this somewhere. And, if it's not known, maybe colloidal Co could be important in some sites (e.g. the hydrothermal vent site, where other metals are known to be enriched in colloidal phases). This is also where labile:dCo is different than throughout the transect.

- Cobalt tends to be scavenged via incorporation into Mn-oxides, in contrast with a metal like Fe, which will more easily form amorphous colloids in the ocean. There is little to no evidence of a significant fraction of colloidal Co in the marine environment (personal communication, Mak Saito). Jessica Fitzsimmons's recent 2022 Ocean Sciences Meeting talk also presented little to no Mn colloids throughout the oceans.

Section 4.6 depth windows: I don't think the depth integration here, giving units of Co per m2, is the best approach. This could be strongly biased by variable depth: at the equator, the depth range integrated here is about 3000 m, while is is about 6000 m at 12 N. I think turning this depth integrated value back into an average (i.e. a depth-weighted average) would give a more meaningful comparison. Likewise, starting this window at 1000 m may be problematic, as it catches the core of the N Pacific OMZ

but is below the OMZ in the subtropical Pacific. These OMZ-specific trends already have two discussion sections, and here the focus seems to be more on the stable dCo in deeper waters. Maybe starting the depth window at 2000 m would better isolate trends in deep water.

- The deep ocean analysis presented in figure 13 e-d has been changed from an integrated inventory to a weighted depth average, which will normalize against variable depths. The depth window of the deep ocean has also been updated to 1500 m, which avoids the deeper NP OMZ.

**Line-specific comments**:

Line 19: Approximately what depth range counts as the "upper ocean"

- The potential density range for this calculation, "($\sigma_0 < 26$)," has been added to the abstract.

Line 20: I suggest using mol:mol ratios rather than mol/l:mol/l

- This change has been made for all ratios except ones that include $O_2$, which remains in units of M:mol $kg^{-1}$. This notation is consistent with previous studies and allows for inter-comparison.

Line 24: Later in the text "potential xs3He" is used rather than "estimated xs3He". It might be better to consistently use only one.

- All uses of "estimated $xs^3He$" have been changed to "potential $xs^3He$" throughout the paper.

Lines 59-60: I suggest "species" instead of "speciation".

- This edit has been made. The sentence now reads "Dissolved Co is present in both a labile, "free" Co(II) species and a ligand-bound, "complexed" species."

Lines 143-147: I think it would be useful here to say slightly more about this offset – specify which method gives higher and which gives lower, and why that might be. Or if it is not consistent, that could also be stated.

- The offset that we see is quite fascinating and broadly interesting to the GEOTRACES and intercalibration communities, but it should be explored as a collaboration between data generators of each type. Generally, we see trends where "labile ICP" dCo falls between electrochemical measurements of labile dCo and total dCo. The purpose of mentioning the difference in this manuscript is to clarify the distinction between the dCo analytical methods for the utilization of the GEOTRACES Intermediate Data Product (IDP) and those who will use it in the future. We have added this last point to the manuscript, and the section now reads:

  "These labile dCo measurements are not necessarily comparable to dCo measurements that are performed using non-electrochemical methods, which are sometimes described as labile. In particular, inductively coupled plasma mass spectrometry (ICP-MS) measurements of dCo where samples are not UV-irradiated prior to analysis gives consistently different results than this electrochemical method, likely due to differences in cobalt complex adsorption and/or exchange kinetics with metal binding resins. As a result, ICP-MS labile Co measurements tend to be somewhat offset compared to electrochemical labile Co measurements (unpublished data). This offset is not yet well understood, but it indicates that the two methods are measuring different

pools of the dCo inventory, which should be considered before comparing dCo values analyzed with different methodologies. The purpose of mentioning the offset in this study is to clarify the distinction between the dCo analytical methods for those who will use this dataset in the future, particularly as part of the GEOTRACES Intermediate Data Product (IDP)."

Lines 236-237 & Lines 536-548: I think small movements in ship position and/or heterogeneity in the hydrothermal plume should be considered. A thousandth of a decimal degree is about 100 m, the same distance quoted as possible due to wire angle. Also, I think Lines 536-548 could be shortened a bit, to simply say that the casts seemed to catch different parts of the plume, and given the depth and local heterogeneity, many different explanations may exist for why they were different.

- This section has been shortened to cut out any lengthy explanations of ship CTD wire angle and position, and now includes a simpler discussion that attributes the possible difference between the two casts to be due to either the natural heterogeneity of the vent field or high wire angles from surface currents. The difference in ship starting position for the CTD casts has also been estimated to be ~15 m, and so the "decimal degree" language has been removed. The paragraph now reads:

  "At the Loihi Seamount, the two CTD casts were completed at the sample station with latitudinal and longitudinal precision to within ~15 m and were deployed within 1.5 hours of each other. However, the casts appeared to sample different sections and/or locations within the hydrothermal plume, as indicated by anomalies in the dFe and dMn data collected from both casts. The GTC cast displayed trace metal (dCo, dFe, dMn) concentrations increasing towards the deepest sample at 1290 m, while trace metals (dFe, dMn) and xs3He from the ODF cast displayed a shallower maximum around 1200–1250 m (Fig. 11a-d). This offset is likely attributable to slight changes in the wire angle of the CTD casts and/or the natural heterogeneity of the hydrothermal plume. The Loihi Seamount sampling station was located near the center of Pele's Pit, a collapsed volcanic pit ~ 200 m deep located towards the southern summit of the seamount. Pele's Pit contains several active hydrothermal vent fields within a radius of ~100 m of the Loihi station, and the casts likely sampled a combined and heterogeneous hydrothermal signature from multiple vents (Clague et al., 2019; Jenkins et al., 2020)."

Lines 365-374: It might be more useful to compare with the Southern Ocean instead of the Atlantic, since the Southern Ocean is more proximally the source of North Pacific deep water.

- dCo scavenging is not a dominant process in the Southern Ocean due to low rates of Mn-oxide formation (Oldham et al. 2021). Instead, we see a more nutrient-like distribution of dCo with little scavenging loss below the mesopelagic. The Atlantic Ocean is a more meaningful comparison to the Pacific when discussing scavenging because it displays similar consistent deep ocean dCo concentrations and experiences similar scavenging processes, which results in a hybrid-type profile.

Line 477: It is a bit ambiguous what the respectively refers to, and from my reading I would think it was slope error and R2 rather than stations 2 and 3 (but what is shown appears to be R2 for stations 2 and 3).

- This sentence has been edited to improve clarity, and the addition of Table 2 describing the numerical results of this analysis also improves the readability of this section. The sentence now reads "In the second (b) and third (c) bins, the dCo:dPO$_4$ slope (86 ± 14 and 86 ± 17 μmol:mol, respectively) is similar to that of the least dense bin (82 ± 3 μmol:mol) but the slope error is increasing and the R2 value is decreasing as bin density increases."

Lines 492-495: The Peru margin is quite a different setting (strong OMZ), and therefore might not be the most directly applicable to this site. Data from an oxic setting might be more useful for a comparison.

- This comparison is not particularly applicable and has been removed.

Lines 499-503: de Carvalho et al (2021) could be a useful reference here.

- While this study is relevant, it does not show a linear relationship between dCo and salinity because its sampling location was so coastal. Instead, it has been added as a reference a few lines later to support the claim that dCo does not act conservatively within river plumes, and so a linear relationship closer to the river endmember would not be expected.

  The section now reads "This estimation assumes riverine dCo is mixing conservatively, which is unlikely to be true since other processes like scavenging and uptake are expected to occur at river mouths and coastal environments (de Carvalho et al., 2021)."

Line 525: Does "Co distributions" refer to dCo or labile dCo or pCo, or all three?

- dCo – this has been clarified and now reads "We observed little evidence that elevated dCo distributions were advected off the shelf…"

Line 528: "Metallic" sounds a bit funny to me.

- "metallic sources" has been changed to "trace metal sources".

Line 588: I'm not sure I understand why a negative correlation is expected. If there is a dCo source from the vent, wouldn't a positive correlation be expected?

- Yes, it would be a positive relationship. This typo has been fixed.

Line 682: It's not clear to me what a volumetric width is. Maybe the calculation can also be described in a bit more detail (perhaps in a supplement, with an example figure)

- The dimensions of the volume used for this calculation has been written to be much clearer to the reader. The description now reads:

  "To predict the effect of this rate of dCo change in the Pacific, the upper ocean (≥ 1000 m) dCo inventory was estimated over the stations within the ETNP and ETSP via trapezoidal integration of dCo station profiles over depth and latitude. The estimated dCo inventories were determined within upper ocean boxes of the Pacific with a width of 1 m, a depth of 1000 m, and a length equal to the distance between the northern-most and southern-most stations affected by the OMZ along a longitude of 152$^o$ W (Table 4)."

Lines 726-746: These two paragraphs are a bit redundant, and can maybe be combined into one paragraph.

- These paragraphs have been combined and a few repetitive sentences cut. The paragraph now reads:

  "One of the most unexpected features in the GP15 dCo transect was the discovery of stable, ligand-bound dCo in the deep interior of the Pacific Ocean, particularly in the North Pacific basin. These higher dCo concentrations do not appear to be due to analytical error or blank correction error, and instead appear to be robust evidence for the presence of complexed dCo below the mesopelagic that has been protected against scavenging. The discovery of elevated dCo in the North Pacific was unexpected, since dCo has been shown to be scavenged along deep thermohaline circulation with deep dCo concentrations decreasing with 14C age (Hawco et al., 2018). When deep GP15 samples (> 3000 m) were paired with their nearest radiocarbon age measurement from the GLODAP database and compared to data from previous GEOTRACES expeditions (Fig. 13a-b; Table 5), many GP15 samples from the North Pacific appeared to deviate from the expected trend of decreasing dCo with water mass age. Deep samples along this transect are estimated to have a conventional radiocarbon age of 918–1645 years, with the intermediate North Pacific containing the oldest waters (>1600 years). The low ventilation of these deep waters can also be seen in the average AOU value at each station below 1000 m depth (Fig. 13e), which displays a steady increase of oxygen utilization with latitude along the transect. Thus, intermediate and deep North Pacific waters were expected to have the lowest, most depleted concentrations of dCo as they are the some of the oldest, least ventilated ocean waters in the world and have had a relatively long timeframe to scavenge their available dCo. While many of the deep GP15 samples, particularly in the South and equatorial Pacific, fell along this predicted trend of slow but steady scavenging within aging water masses, many samples from the North Pacific were relatively high in dCo despite their age. This finding suggests a source of dCo to the deep North Pacific, superimposing a regional source process on top of a global circulation scavenging process."

Lines 806-808: It is probably both of these, and not just one or another, as the rest of the discussion section and figures identify specific examples of these two. In that respect, I think it is useful to have this short discussion section addressing the models, but I think it could be streamlined a bit. It is hard to have a detailed discussion of the differences between models and this dataset without explaining the model in more detail (which isn't the purpose of this manuscript). I recommend spending less time discussing detailed differences between the model and the new data, and center the discussion on highlighting how this dataset can aid future models improve based on the new constraints on the Co cycle discovered here. This mostly applies to the final paragraph, for example where the initial presentation suggests that dCo is more scavenged in the Pacific than expected (in contradiction to the earlier discussion), but rather it is just that the model used a different parameterization than these new data would suggest (i.e. opportunity to improve the model).

- The text was never intended to imply that, between options (1) and (2) presented in line 807-8, only one explanation was true. To clarify this point, the sentence now reads: "The main differences between the observed and model dCo values can be attributed to both (1)… and (2)…"

The discussion of model data as it compares to the observed data has been updated to reflect more on the future of the model and how this comparison will be used to improve upon future iterations of the model. The final paragraph of this section now reads:

"Recognizing inconsistencies between expected model dCo distributions and observed transect dCo distributions is an effective way to test and improve upon the biogeochemical model's parameters. The parameterization of ligand-bound cobalt in the deep Pacific, for example, could be improved by decreasing the assumed concentration of inert, organically bound dCo in deep water masses. It should be noted that the biogeochemical model was optimized for the global ocean Co cycle, and not specifically for the Pacific Ocean, which likely contextualizes much of the offset between the observed and model transects, particularly for inconsistencies within the $O_2$ and $PO_4$ distributions. The model also did not incorporate a hydrothermal vent source to the Pacific, which we concur is an appropriate omission on a basin scale. Future iterations of the Co biogeochemical model will consider the successes and inconsistencies presented here and continue to improve upon our conceptualization and parameterization of the marine cobalt cycle."

Lines 860-868: I think it is reasonable to include this, but the transitions to and from this section are a bit abrupt, and this topic wasn't addressed anywhere else in the text. Maybe it could be moved or somehow incorporated better, so that it doesn't break the flow of points before and after (as built up by the rest of the manuscript).

- This section has been expanded and more of a transition from the general biogeochemical cycle of cobalt to the mining implications of such studies has been added so that it more clearly fits into this conclusion section. The paragraph now reads:

"This study improves our understanding of dCo dynamics in the Pacific water column, but it should be mentioned that many are interested in Pacific Co biogeochemical cycling as it relates to Co scavenging and mineralization along the seafloor and the emerging deep sea mining industry. When dCo is scavenged and removed from the water column in the form of pCo incorporated into Mn-Oxide particles, it sinks and forms vast regions of ferro-manganese oxide crusts embedded with metal-rich nodules that accrete slowly along the seafloor (Aplin and Cronan, 1985). These nodules are formed from Mn oxide deposition over millennia and are rich in Co, Ni, Cd, Zn and rare earth elements (REEs) (Cameron et al., 1981; Hein et al., 2013). Such deposits are of interest to many because of the potential for deep-sea mining in the region, especially in the Clarion-Clipperton Zone of the central North Pacific basin. Here, international mining operations to extract Fe-Mn nodules along the seafloor could being within the decade. The value of Co ore, which is currently only mined terrestrially, has been increasing in demand over the past two decades due to the metal's use in personal electronic devices and sustainable energy solutions like electric vehicle batteries, and many believe that mining the vast, concentrated deposits of Co on the Pacific seafloor could alleviate this demand (Hein et al., 2013). However, deep-sea mining and the potential heavy-metal rich sediment plumes it could create may have serious ramifications to the relatively fragile and unexplored benthic ecosystem in the region (Drazen et al., 2020; Fuchida et al., 2017; Sharma, 2011). Although the technology and facilities required to mine the deep sea have proven to be both complicated and expensive (Cameron et al., 1981), the potential new source of Co ore could, in theory, help support widespread adoption of electric vehicle fleets (Hein et al., 2013)."

General comment on figures: It would be nice if the figures could be a bit more internally consistent, and a bit more easy to differentiate. A generally consistent color scheme is used throughout the figures, but this is applied inconsistently to the parameters plotted. For example, Figure 5 uses teal to mean dissolved Co while the other figures use blue to signify this, and teal to indicate labile cobalt.

- Although the figure color schemes were never intended to be consistent across all figures, this is a great idea and will help with reader comprehension. The figure color scheme now prioritizes navy blue circles for dCo and teal squares for labile dCo where possible.

Figures 3 & 5: I think adding station numbers above the top panel would be helpful.

- Station numbers have been added to figures 3 and 5.

Figure 4: It would be good to include visual color legend somewhere, even if it is a bit intuitive. Also, I find this figure a bit hard to read. Maybe the figure would be easier to read if the surface box was somehow expanded (either it took more space, or it covered 0-500 m instead of 0-1000). Also, it would be nice if the scaling for the upper 1000 m was the same for the deep stations as it is for the partial stations (7, 9, 13, 15, 17, 20, 22, 34, 36, 38).

- Figure 4 has been reconfigured to be easier to read, with more space between profiles and a visual legend. The shallow profile boxes have also been expanded to match the size and scale of the demi-station profile boxes for consistency.

Figure 5: It would be nice to include the latitudinal zones here, as with figure 3. It also seems like the Loihi Seamount is missing here. Also, the order of panels (PO4 and O2 specifically) is reversed compared to Figure 3. Finally, maybe this could be combined with Figure 3 to make an 8 panel figure.

- Oceanographic regions from figure 3 have been added to figure 5, and the order of the PO4 and O2 panel has been switched to match figure 5. The Loihi Seamount station was present in the original figure, it just contains few CTD depths in the surface ocean (above 1000 m). The Loihi and Puna Ridge stations have been clearly marked.

Figure 6: The y scaling is different in each of these three panels. Maybe the bottom two could use the same range.

- The y-axes of the bottom two panels now use the same scale, and the marker colors and shapes have been updated to be more consistent with the color schemes of other figures.

Figure 7: I don't think the 1:1 line actually shows a 1:1 relationship. It looks like 120 pM labile Co is less than 100 pM total dCo on this line.

- Thank you for catching this – there was a typo in the code for the 1:1 line. The line is now correct.

Figure 9: Panel J is a great addition! Maybe the other panels could be improved by writing the approximate depth covered by this isopycnal range in parenthesis after the isopycnals. This can be re-created by going back and forth between the panels, but having it written would be nice.

- A table (Table 2) has been added with summary information of the isopycnal analysis results, including the average depth. A rounding error in the annotations of Figure 9 was also corrected.

Figure 10: Please use different colors for Mn, Ti and Al compared to each other and what is used for Co. Also, the ranges of LPT and SPT are different, but this isn't obvious and it gives the impression that LPT is approximately the same as SPT. If it is necessary to use different ranges, please make this more clear, and maybe differentiate the ranges so the data don't overlap.

- The colors of panels (d), (e) and (f) have been changed, and the ranges of the secondary x-axes have been modified so they no longer appear to intercept with the primary x-axis. A note in the figure caption has also been added to bring attention to the changes in x-axis range.

Figure 11: Here dCo is presented as purple (panel A), compared to blue in most of the other plots.

- The colors of this figure have been updated such that the dCo profile is navy, dFe and dFe-derived values are purple, and dMn and dMn-derived values are teal.

Figure 13: I don't see corrected pCo on panel C. The color choice here could create some confusion with labile dCo. Also, it might be nice if Panel D didn't use the Co color scheme, and if the different ranges could be somehow better differentiated (as with Figure 10). As it is now, it gives the initial impression that LPT POC = SPT POC most of the time.

- The colors of this figure have been updated to be consistent with colors used for dCo, LPT POC, and SPT POC throughout the paper, and a note in the figure caption has been add to draw attention to the order of magnitude difference between LPT and SPT POC in panel d.

**References:**

de Carvalho, L.M., Hollister, A.P., Trindade, C., Gledhill, M. and Koschinsky, A., 2021. Distribution and size fractionation of nickel and cobalt species along the Amazon estuary and mixing plume. *Marine Chemistry*, *236*, p.104019.

Jenkins, W.J., Hatta, M., Fitzsimmons, J.N., Schlitzer, R., Lanning, N.T., Shiller, A., Buckley, N.R., German, C.R., Lott III, D.E., Weiss, G. and Whitmore, L., 2020. An intermediate-depth source of hydrothermal 3He and dissolved iron in the North Pacific. *Earth and Planetary Science Letters*, *539*, p.116223.

Martin, J.H., Gordon, R.M., Fitzwater, S. and Broenkow, W.W., 1989. VERTEX: phytoplankton/iron studies in the Gulf of Alaska. *Deep Sea Research Part A. Oceanographic Research Papers*, *36*(5), pp.649-680.

Ohnemus, D.C., Torrie, R. and Twining, B.S., 2019. Exposing the distributions and elemental associations of scavenged particulate phases in the ocean using basin-scale multi-element data sets. *Global Biogeochemical Cycles*, *33*(6), pp.725-748.

Sim, N. and Orians, K.J., 2019. Annual variability of dissolved manganese in Northeast Pacific along Line-P: 2010–2013. *Marine Chemistry*, *216*, p.103702.

Appendix A

[Figure]

**Figure A1: Interpretations of the dCo:O₂ relationship in the North Pacific (a-d; stations 3-18), ETNP (e-h; stations 20-25), and ETSP (i-l; stations 29-34). Subplots a, e and i show dCo vs. O₂ where depth ≥ 500 m in the North Pacific and depth ≥ 200 m in the ETNP and ETSP. Subplots b, f and j include shallower samples above the 500 m depth threshold in the North Pacific and 200 m depth threshold in the ETNP and ETSP. Subplots c, g and k separate the dCo vs. O₂ correlations by O₂ concentration, with lower [O₂] samples (< 50 µmol kg⁻¹) displaying more negative slopes than samples where 50 ≤ [O₂] ≤ 150 µmol kg⁻¹. This trend suggests low-[O₂] samples, which tend to occur at shallower depths, are affected by more vertical processes like remineralization and scavenging at the boundaries of the OMZ core, compared to mid-[O₂] samples, which are affected by more horizontal processes of advection and mixing. Two-way regression statistics are given in Table S1 below. Subplots d, h and l include labile dCo concentrations and suggest that labile dCo might be more prevalent in the low-[O₂] samples, and could be scavenged in more oxic waters. The labile dCo concentrations appear to be affected by additional processes in the North Pacific.**

**Table A1: Two-way regression statistics for the dCo vs. O2 relationship in the three Pacific OMZs, separated by $O_2$ concentration (shown in Fig. S1 c,g,k). In the North Pacific OMZ, samples at and below 500 m depth were included in the regression, and in the ETNP and ETSP, samples at and below 200 m depth were included. Note that $R^2$ values were particularly low in the North Pacific and low-$[O_2]$ ETNP regressions.**

| OMZ | Stations Included | Latitudinal Range | $[O_2] < 50$ μmol kg$^{-1}$ | | | $50 \leq [O_2] \leq 200$ μmol kg$^{-1}$ | | |
|---|---|---|---|---|---|---|---|---|
| | | | dCo:$O_2$ [μM:mol kg$^{-1}$] | $R^2$ | n | dCo:$O_2$ [μM:mol kg$^{-1}$] | $R^2$ | n |
| North Pacific | 3 - 18 | 55º to 22º N | -6.4 ± 4.5 | 0.03 | 75 | -0.23 ± 0.12 | 0.05 | 69 |
| ETNP | 20 - 25 | 14.25º to 5º N | -2.8 ± 1.8 | 0.09 | 26 | -0.58 ± 0.10 | 0.54 | 28 |
| ETSP | 29 - 34 | 0º N to 7.5º S | -2.1 ± 0.6 | 0.47 | 13 | -0.63 ± 0.07 | 0.65 | 38 |